# SALAAD: Sparse And Low-Rank Adaptation via ADMM for Large Language Model Inference

**Hao Ma** [1 2]  **Melis Ilayda Bal** [2 3]  **Liang Zhang** [1 2]  **Bingcong Li** [1]  **Niao He** [1]  **Melanie Zeilinger** [1]
**Michael Muehlebach** [2]

## Abstract

Modern large language models are increasingly deployed under compute and memory constraints, making flexible control of model capacity a central challenge. While sparse and low-rank structures naturally trade off capacity and performance, existing approaches often rely on heuristic designs that ignore layer and matrix heterogeneity or require model-specific architectural modifications. We propose SALAAD, a plug-and-play framework applicable to different model architectures that induces sparse and low-rank structures during training. By formulating structured weight learning under an augmented Lagrangian framework and introducing an adaptive controller that dynamically balances the training loss and structural constraints, SALAAD preserves the stability of standard training dynamics while enabling explicit control over the evolution of effective model capacity during training. Experiments across model scales show that SALAAD substantially reduces memory consumption during deployment while achieving performance comparable to ad-hoc methods. Moreover, a single training run yields a continuous spectrum of model capacities, enabling smooth and elastic deployment across diverse memory budgets without the need for re-training.

## 1. Introduction

Large language models (LLMs) have achieved remarkable performance across a wide range of tasks (Minaee et al., 2025; Zhao et al., 2025; Zhou et al., 2023; Liu et al., 2023; Dong et al., 2024; Huang & Chang, 2023). However, their rapidly increasing scale poses significant challenges for practical deployment across heterogeneous compute and memory budgets, spanning cloud GPU clusters and resource-constrained edge devices (Zheng et al., 2025; Lin et al., 2024; Zhang et al., 2024b). For example, server-grade GPUs such as `NVIDIA A100`/`H100` offer up to 80 GB of device memory, whereas consumer or edge platforms, such as the `Raspberry Pi 5` and `NVIDIA Jetson` systems, are typically limited to less than 16 GB, resulting in orders-of-magnitude differences in feasible model capacity. Despite this diversity, state-of-the-art LLMs are usually trained and released at only a few fixed scales, forcing practitioners to trade off resource efficiency against performance (Brown et al., 2020; Ruan et al., 2024; Grattafiori et al., 2024; Touvron et al., 2023; Team et al., 2024a;b; DeepSeek-AI et al., 2025a;b).

To mitigate the deployment challenges mentioned above, a broad range of model compression and efficiency techniques have been explored, including quantization (Dettmers et al., 2022; Frantar & Alistarh, 2022; Egiazarian et al., 2024; Lin et al., 2024; Tseng et al., 2024), pruning (Xia et al., 2022; Ma et al., 2023; Zhang et al., 2024a; Xia et al., 2024), knowledge distillation (Wang et al., 2021; Sun et al., 2020; Jiao et al., 2020; Gu et al., 2025; Liang et al., 2023), and model approximation (Hsu et al., 2022; Chen et al., 2021; Li et al., 2023; Saha et al., 2023). Among these, approximation-based methods reduce effective model capacity by replacing dense parameters with structured representations, enabling a principled trade-off between expressiveness and memory efficiency. In this work, we focus on sparse and low-rank (SLR) approximation, a compact yet expressive parameterization that has been extensively studied in LLMs (Mo et al., 2024; Liu et al., 2025; Han et al., 2024; Li et al., 2025).

Approximation can be applied either post hoc (Yu et al., 2017; Candès et al., 2011; Lin et al., 2010; Hu et al., 2021) or during training as part of the optimization process (Han et al., 2024; Li et al., 2025). A promising class of approximation-based methods imposes SLR structure to reduce effective model capacity while preserving expressiveness (Candès et al., 2011; Lin et al., 2010). While such parameterizations can be theoretically expressive, for exam-

---

[1]ETH Zurich [2]Max Planck Institute for Intelligent Systems [3]École polytechnique fédérale de Lausanne (EPFL). Correspondence to: Hao Ma <haomah@ethz.ch>.

*Proceedings of the 43rd International Conference on Machine Learning*, Seoul, South Korea. PMLR 306, 2026. Copyright 2026 by the author(s).

ple, Proposition 1 in SLTrain (Han et al., 2024) shows that a low-rank matrix augmented with random sparsity is full-rank with high probability, this guarantee critically depends on strong randomness and independence assumptions on the sparse support. These assumptions are unlikely to hold when SLR approximation is applied post hoc to fully trained dense models, whose weights exhibit structured correlations across layers, violating these assumptions and often leading to performance degradation (see Appendix A). Consequently, post-hoc SLR approximation is fundamentally misaligned with the conditions under which such theoretical guarantees apply. Recent works have explored training-time approximation by incorporating SLR structure into pretraining, with representative examples including SLTrain (Han et al., 2024) and LOST (Li et al., 2025). However, these methods rely on heuristic, layer-agnostic structure schedules (e.g., fixed truncation ratios or sparsity patterns), ignoring spectral heterogeneity across blocks[1] and often coupling the approach to specific parameterizations or architectures. As a result, a principled and training-compatible framework that generalizes across models and enables flexible, deployment-time capacity control remains lacking.

In this paper, we propose **SALAAD** (Sparse And Low-Rank Adaptation via ADMM), a plug-and-play framework for inducing adaptive SLR structure during pretraining. SALAAD formulates structured approximation as an optimizer-side objective within an augmented Lagrangian framework, enabling seamless integration into standard training without architectural modification. To avoid rigid or hand-crafted structural constraints, we introduce an adaptive control that dynamically regulates the strength of SLR induction across components during training, allowing gradual structure induction while preserving training stability. The resulting SLR decomposition naturally supports elastic deployment under varying memory constraints: a single pretrained model can continuously adjust its effective rank or sparsity post hoc to meet diverse memory budgets, without retraining or architectural modification. Unlike architecture-dependent approaches such as MatFormer (Kudugunta et al., 2023) and Flextron (Cai et al., 2024), which rely on nested or routed architectures and support only a discrete set of submodels, SALAAD operates entirely in weight space and enables fine-grained, deployment-aware capacity scaling. We further observe that under adaptive SLR induction, embedding layers naturally exhibit stable SLR structure without noticeably affecting training dynamics, indicating that they are inherently SLR-inducible within this framework. Our contributions are summarized as follows:

---

[1] Throughout this paper, a **block** refers to a distinct linear mapping within a Transformer architecture, such as the $X_q$, $X_k$, $X_v$, $X_o$ projections in attention or the up/gate/down projections in MLPs. When explicitly noted, the term may also include non-standard linear mappings such as the embedding matrix or the LM head.

- We address the challenge of inducing SLR structure during LLM pretraining without destabilizing optimization or modifying model architectures. We introduce **SALAAD**, a training-compatible, plug-and-play framework that jointly optimizes dense and structured surrogate models under a controlled approximation, and applies to Transformer-style architectures.
- We alleviate the need for heuristic, layer-agnostic structure schedules in existing training-time approximation methods. We introduce an I(ntegral)-controller that adaptively regulates rank and sparsity at the block level during training, eliminating hand-crafted truncation ratios and sparsity patterns. This mechanism reduces structural hyperparameters to a single penalty coefficient and enables different Transformer blocks to acquire appropriate SLR structure, thereby allowing previously hard-to-regularize components, such as embedding layers, to be safely incorporated into training-time SLR induction.
- We address the lack of continuous, architecture-preserving capacity control in existing efficient LLM deployment methods. We introduce a homomorphic parameter allocation (HPA) strategy that adjusts model capacity continuously in weight space without the need for retraining or architectural modification. As a result, a single pretrained checkpoint supports fine-grained deployment across diverse memory and compute budgets with predictable performance-capacity trade-offs.
- We provide a pretraining-time empirical characterization of how different LLM components respond to SLR induction, revealing a pronounced SLR regularization asymmetry across submodules. Specifically, embedding layers exhibit **benign and stable SLR behavior** under adaptive induction, often without degrading perplexity, whereas the language modeling (LM) head cannot be SLR-induced without performance loss. This asymmetry exposes previously unexplored structural properties of major LLM components and underscores the importance of component-aware SLR induction during pretraining. To our knowledge, this asymmetry has not been characterized in prior pretraining research.

## 2. Related Work

Parameter-efficient fine-tuning methods such as LoRA (Hu et al., 2021; Lialin et al., 2023; Liu et al., 2024; Lion et al., 2025) introduce low-rank adaptations after pretraining to reduce the cost of downstream task adaptation. While effective, these approaches operate in a post-training regime and require additional optimization. In contrast, another line of work aims to induce SLR structures during pretraining, differing primarily in how structural constraints are imposed. GaLore (Zhao et al., 2024) reduces training memory by projecting gradients onto low-rank subspaces but yields dense models at inference. LORO (Mo et al., 2024) directly pa-

rameterizes linear layers as fixed-rank operators, requiring manual rank specification, while CoLA (Liu et al., 2025) enforces low-rank structure via bottleneck operators but relies on architectural modifications. Among these methods, SLTrain (Han et al., 2024) and LOST (Li et al., 2025) are most closely related to our work, as they train models with pre-parameterized SLR structures throughout pretraining; however, their rank and sparsity patterns are fixed prior to training, require nontrivial modifications to linear layers or the training pipeline, and do not account for block-wise heterogeneity, limiting their plug-and-play applicability.

More broadly, these methods focus on structured optimization during pretraining, whereas our approach targets deployment-time flexibility by producing a structured surrogate model that supports continuous, architecture-preserving capacity control after training. This perspective complements elastic inference frameworks such as Flextron (Cai et al., 2024) and MatFormer (Kudugunta et al., 2023), which rely on discrete architectural choices or predefined submodels. By operating directly in weight space, SALAAD enables fine-grained adaptation under varying resource constraints without architectural modification.

## 3. Problem Formulation

Let $\pi$ denote a neural network parameterized by weights $\{X_i\}_{i=1}^N$, where $N$ is the number of selected blocks. Each block $X_i \in \mathcal{X}_i \subset \mathbb{R}^{n_i \times m_i}$ is decomposed into a low-rank component $L_i \in \mathcal{L}_i$ and a sparse component $S_i \in \mathcal{S}_i$, such that $X_i = L_i + S_i$, where $\mathcal{X}_i$, $\mathcal{L}_i$, and $\mathcal{S}_i$ are assumed to be closed convex sets. Since blocks are treated independently, we omit the index $i$ and focus on a single block $X \in \mathcal{X}$ with components $L \in \mathcal{L}$ and $S \in \mathcal{S}$. The low-rank component captures the dominant structure of each block, while the sparse component preserves fine-grained residual variations beyond the low-rank approximation. Inspired by Robust Principal Component Analysis (RPCA) (Candès et al., 2011; Lin et al., 2010; Hong et al., 2015; Wang et al., 2014), we formulate the optimization problem as follows:

$$
\begin{aligned}
\min_{X \in \mathcal{X}} \quad & \ell(X) + \alpha |L|_* + \beta |S|_1 \\
\text{s.t.} \quad & X = L + S,
\end{aligned}
\tag{1}
$$

where $\ell : \mathcal{X} \to \mathbb{R}$ denotes the task-specific training loss. The nuclear norm $|\cdot|_*$ and the element-wise $\ell_1$-norm $|\cdot|_1$ serve as tractable convex surrogates for rank and sparsity, respectively, with $\alpha > 0$ and $\beta > 0$ controlling the strength of the low-rank and sparse regularization. Solving (1) yields a weight matrix $X$ that admits an effective low-rank and sparse decomposition $X = L + S$. This decomposition enables compression during training by reducing the number of effective parameters while preserving the expressive capacity needed to maintain predictive performance.

## 4. Methodology

Appendix B provides a visual overview of the pretraining and deployment stages of SALAAD.

### 4.1. ADMM for SLR Decomposition

We propose the Alternating Direction Method of Multipliers (ADMM) algorithm (Hong et al., 2015; Wang et al., 2014; Candès et al., 2011; Lin et al., 2010; Ouyang et al., 2013) for solving (1). We first define the augmented Lagrangian function for the problem as follows:

$$
\begin{aligned}
\mathcal{G}_\rho(X, L, S, Y) = {} & \ell(X) + \alpha |L|_* + \beta |S|_1 \\
& + \frac{\rho}{2}\left|X - L - S + \frac{Y}{\rho}\right|_{\mathrm{F}}^2 - \frac{1}{2\rho}|Y|_{\mathrm{F}}^2,
\end{aligned}
$$

where $Y \in \mathbb{R}^{n \times m}$ is the dual variable associated with the equality constraint, $\rho > 0$ is the penalty parameter, and $|\cdot|_{\mathrm{F}}$ denotes the Frobenius norm. The ADMM algorithm then iteratively updates the primal variables $(X, L, S)$ and the dual variable $Y$ for $k \geq 0$ as follows:

$$
X_{k+1} \doteq \arg\min_{X \in \mathcal{X}} \mathcal{G}_\rho(X, L_k, S_k, Y_k),
\tag{2}
$$

$$
L_{k+1} \doteq \arg\min_{L \in \mathcal{L}} \mathcal{G}_\rho(X_{k+1}, L, S_k, Y_k),
\tag{3}
$$

$$
S_{k+1} \doteq \arg\min_{S \in \mathcal{S}} \mathcal{G}_\rho(X_{k+1}, L_{k+1}, S, Y_k),
\tag{4}
$$

$$
Y_{k+1} \doteq Y_k + \rho(X_{k+1} - L_{k+1} - S_{k+1}).
\tag{5}
$$

The updates for $L$ and $S$ in (3) and (4) can be computed in closed form using proximal operators. Specifically, the update for $L$ is given by:

$$
L_{k+1} = \operatorname{prox}_{\frac{\alpha}{\rho}|\cdot|_*}(X_{k+1} - S_k + {}^{Y_k}\!/_\rho),
$$

where the proximal operator $\operatorname{prox}_{\tau|\cdot|_*}$ is defined as:

$$
\operatorname{prox}_{\tau|\cdot|_*}(Z) = U\operatorname{diag}\left(\{(\sigma_i - \tau)_+\}\right)V^\top,
$$

with $Z = U\operatorname{diag}(\{\sigma_i\})V^\top$ being the singular value decomposition (SVD), and $(x)_+ \doteq \max\{x, 0\}$. Similarly, the update for $S$ is given by:

$$
S_{k+1} = \operatorname{prox}_{\frac{\beta}{\rho}|\cdot|_1}(X_{k+1} - L_{k+1} + {}^{Y_k}\!/_\rho),
$$

where the proximal operator $\operatorname{prox}_{\tau|\cdot|_1}$ is defined in an element-wise manner as:

$$
\left[\operatorname{prox}_{\tau|\cdot|_1}(Z)\right]_{ij} = \operatorname{sign}(Z_{ij})(|Z_{ij}| - \tau)_+.
$$

Due to the highly nonconvex nature of neural networks, the $X$-update in (2) does not admit a closed-form solution and is instead approximated using a gradient-based step at

**Algorithm 1** SALAAD (block-wise)
> **Initialization:** $X$, $L$, $S$ and $Y$; $\alpha$, $\beta$ and $\rho$
> **while** TRUE **do**
>     $X_0 \leftarrow X$
>     /* GUIDED LEARNING PROCESS */
>     **for** $k$ in $1, \ldots, K$ **do**
>         Sample a minibatch of data
>         Do the backpropagation for $\ell_c = \ell + \ell_\rho$ to get $X_k$
>     **end for**
>     $X \leftarrow X_K, L_0 \leftarrow L, S_0 \leftarrow S, Y_0 \leftarrow Y$
>     /* SPARSE AND LOW-RANK ADAPTATION */
>     **for** $j$ in $1, \ldots, J$ **do**
>         $U, s, V^\top = \text{SVD}(X - S_{j-1} + \frac{Y_{j-1}}{\rho})$
>         $L_j = U \mathcal{T}_{\alpha/\rho}(s) V^\top$
>         $S_j = \mathcal{T}_{\beta/\rho}(X - L_j + Y_{j-1}/\rho)$
>         $Y_j = Y_{j-1} + \rho(X - L_j - S_j)$
>     **end for**
>     $L \leftarrow L_J, S \leftarrow S_J, Y \leftarrow Y_J$
>     /* REGULARIZATION PARAMETERS UPDATE */
>     $\alpha, \beta \leftarrow \text{I-CONTROLLER}(\alpha, \beta, L, S)$
> **end while**

each iteration. However, for LLM pretraining, solving the $X$-update by repeatedly traversing the full dataset is impractical. To address this, we introduce Algorithm 1, which reformulates classical ADMM into a two-stage optimization procedure suitable for large-scale training. We note that $\mathcal{T}$ denotes the soft thresholding. In the first-stage optimization, we randomly sample a minibatch of data and update $X$ with a gradient step on the coupled loss $\ell_c$ using any existing optimizer, where

$$\ell_c(X) \doteq \ell(X) + \underbrace{\frac{\rho}{2}|X - L - S + Y/\rho|_F^2}_{\ell_\rho(X)}. \quad (6)$$

In (6), $\ell_\rho$ not only serves as a soft constraint that limits the magnitude of updates to $X$, but also continuously guides $X$ toward a manifold exhibiting SLR structure during the optimization process.

After updating $X$, we enter a second-stage optimization, where the closed-form updates in (3)-(5) are applied to recover the SLR structure of $X$. Algorithm 1 thus produces two sets of weights: the original parameters $X$ and a structured surrogate $\widehat{X} = L + S$. With an appropriate choice of $\rho$, the discrepancy $|X - \widehat{X}|_F$ remains bounded during training (see Appendix F). While $\widehat{X}$ is SLR by construction, $X$ itself is not explicitly required to be SLR.

This two-stage design decouples the original ADMM into two complementary processes. In the first-stage optimization, the coupled loss $\ell_c$ encourages $X$ to gradually approach the SLR submanifold in parameter space. In the second-stage optimization, an explicit SLR representation

of $X$ is recovered via the proximal operators in (3)-(5). Together, these steps yield a scalable and practical ADMM variant suitable for LLM pretraining. The memory and computational cost analysis of Algorithm 1 is provided in Appendix C.

### 4.2. I-Controller for Adaptive Regularization

In large-scale LLM training, hyperparameters that follow predictable scaling laws are critical for robustness at scale. Algorithm 1 is a plug-and-play procedure layered on top of an existing optimizer, and introduces two classes of hyperparameters: i) those inherited from the base optimizer, and ii) the SALAAD-specific hyperparameters, namely the *block-wise* penalty coefficient $\rho$ and the *block-wise* SLR thresholding levels $(\alpha, \beta)$.

For the first class, extensive experiments show that the inductive term $\ell_\rho$ in (6) does not interfere with the behavior of the underlying optimizer. As a result, standard optimizer hyperparameters (e.g., learning rate schedules and momentum coefficients) can be reused without modification.

Naively, the second class would require separate $(\rho, \alpha, \beta)$ for each block. We instead introduce an I-controller that reduces this space to a single global coefficient $\rho$, while adaptively adjusting block-wise rank and sparsity thresholds. This design enables block-specific SLR evolution according to functional role, eliminates per-block manual tuning, and empirically yields a consistent scaling law for $\rho$, allowing SALAAD to scale to models with hundreds of blocks and billions of parameters.

Specifically in our work, the I-controller adjusts $\alpha$ and $\beta$ as follows:

$$\alpha \leftarrow \alpha + \rho(\Gamma_L^\gamma - \widehat{\Gamma})\Delta\alpha, \ \beta \leftarrow \beta + \rho(\Upsilon_S - \widehat{\Upsilon})\Delta\beta,$$

where $\Delta\alpha, \Delta\beta > 0$ are small step sizes, $\Gamma_L^\gamma$ denotes the effective rank ratio of $L$ under energy coverage[2], and $\Upsilon_S$ denotes the density of $S$, where the effective rank ratio under energy coverage is defined as follows:

**Definition 4.1** (Effective Rank Ratio under Energy Coverage)**.** Given a matrix $L \in \mathbb{R}^{n \times m}$ with singular values $\sigma_1 \geq \sigma_2 \geq \cdots \geq \sigma_{\min\{n,m\}} \geq 0$, the effective rank of $L$ under energy coverage $\gamma \in (0, 1]$ is defined as:

$$\Gamma_L^\gamma \doteq \frac{\min\left\{k : \sum_{i=1}^k \sigma_i / \sum_{j=1}^{\min\{n,m\}} \sigma_j \geq \gamma\right\}}{\min\{n, m\}}.$$

Additionally, let $\widehat{\Gamma} \in [0, 1]$ and $\widehat{\Upsilon} \in [0, 1]$ denote the target effective rank under energy coverage and target density, respectively, which are designed by the user based on deployment requirements. These hyperparameters are easier

---

[2]We use $\gamma = 0.999$ for numerical stability.

to choose in practice, as they directly correspond to global compression objectives rather than layer-specific structural choices. Under the adaptive control mechanism, $\widehat{\Gamma}$ and $\widehat{\Upsilon}$ should be slightly below the desired deployment targets.

By fixing $\widehat{\Gamma}$, $\widehat{\Upsilon}$, and $(\Delta\alpha, \Delta\beta)$, the behavior of SALAAD during training is governed by a single hyperparameter, namely the block-wise penalty coefficient $\rho$. In practice, tuning $\rho$ alone is sufficient to ensure stable and effective convergence of Algorithm 1 (see Section 5.3 and Appendix I). Intuitively, $\rho$ controls the relative strength of the SLR-inducing penalty $\ell_\rho$ in (6) with respect to the task loss $\ell$. Larger values of $\rho$ enforce stronger SLR structure, driving the effective rank and density closer to their targets, but may degrade $\ell$ if overly aggressive; conversely, smaller $\rho$ favors task loss minimization but weakens the induced SLR structure. Thus, $\rho$ governs the fundamental trade-off between task performance and structural fidelity.

Motivated by this observation and inspired by (Lin et al., 2010), we adopt the following practical scaling law for LLM training:

$$\rho \propto \frac{1}{N\sqrt{n \times m}}, \tag{7}$$

where $N$ is the number of selected blocks in the model, and $n$ and $m$ are the dimensions of the weight matrix $X$.

This scaling form ensures consistency across model depth and parameter dimensionality. The factor $1/\sqrt{nm}$ normalizes the penalty by the typical scale of the weight matrix, whose Frobenius norm empirically scales as $\mathcal{O}(\sqrt{nm})$, while $1/N$ compensates for the accumulation of regularization across blocks, preventing the total penalty from growing linearly with model depth. In practice, we tune $\rho$ on 60M and 130M models to determine the proportionality constant, and then fix it when scaling to larger models. This enables reuse of a single hyperparameter setting across model sizes with minor per-scale tuning. In practice, we use small models to calibrate the proportionality constant in (7) and select the controller step sizes. The same target rank and density ratios are then reused across model scales, while the ADMM update frequency is chosen to balance structural tracking quality and training cost.

### 4.3. Homomorphic Parameter Allocation Strategy

After SALAAD training, the structured surrogate $\{\widehat{X}_i\}_{i=1}^N$ typically contains more SLR units than required by the nominal target structure, as the block-wise I-controller adapts to training dynamics rather than enforcing strict convergence to prescribed ranks and densities. This surplus provides additional degrees of freedom for post-hoc compression.

We formalize post-hoc truncation as a budgeted selection problem. Let $\mathcal{U} = \{u_1, \ldots, u_M\}$ denote all removable units, where each $u$ corresponds to either a singular value

from $L_i$ or a nonzero entry from $S_i$, with $i = 1, \ldots, N$. Removing a unit $u$ deletes $c(u) \in \mathbb{N}_+$ parameters, and its importance is defined as the expected loss increase

$$I(u) = \mathbb{E}\left[\ell\big(\{\widehat{X}_i\}_{i=1}^N \setminus u\big) - \ell\big(\{\widehat{X}_i\}_{i=1}^N\big)\right],$$

which captures the functional impact of removing each unit.

Assuming independent effects across units, the truncation under a given parameter budget $C \in \mathbb{N}$ can be written as

$$\min_{\mathcal{U}' \in \mathcal{U}} \sum_{u \in \mathcal{U}'} I(u) \quad \text{s.t.} \quad \sum_{u \in \mathcal{U}'} c(u) \geq C. \tag{8}$$

Solving the above optimization problem amounts to selecting a subset of removable units that minimizes the total importance while removing at least $C$ parameters. We notice that (8) is a budgeted discrete optimization problem that is intractable in practice since $I(u)$ is not directly computable.

To address this, we introduce a homomorphic parameter allocation (HPA) strategy as an efficient greedy approximation. First, we assume that unit importance $I(u)$ is proportional to its magnitude $I(u) \propto |u|$, which is well motivated for singular values and a reasonable proxy for sparse entries. Second, we assume structural homomorphism across blocks, in that the SLR components in all blocks are scaled according to a shared global ratio, rather than being individually tuned based on block-specific sensitivity.

Under these assumptions, we introduce a mixing coefficient $\kappa \in [0, 1]$ to split the total budget $C$ between SLR components. Specifically, $\kappa C$ parameters are removed from $\{L_i\}_{i=1}^N$ and $(1 - \kappa) C$ from $\{S_i\}_{i=1}^N$. The resulting global scaling ratios[3] are

$$\phi_L \doteq \frac{\kappa C}{C_L}, \quad \phi_S \doteq \frac{(1 - \kappa) C}{C_S}, \tag{9}$$

where $C_L$ and $C_S$ denote the total removable parameters in SLR components, respectively. These ratios are applied uniformly across blocks.

*Remark* 4.2 (HPA preserves learned heterogeneity). The global ratios $\phi_L$ and $\phi_S$ in HPA are applied to the learned low-rank and sparse components, rather than imposing a uniform rank or sparsity level on all blocks. Since different blocks develop distinct effective ranks and sparse densities during training, proportional truncation preserves these relative differences while satisfying a global parameter budget. Thus, HPA is a greedy deployment-time approximation for continuous budget adaptation without retraining.

For each block, truncation is implemented by sorting singular values and sparse entries by magnitude and removing

---

[3] For simplicity, we assume that $\phi_L \leq 1$ and $\phi_S \leq 1$. If either bound is violated, the surplus budget is reassigned to the other component. Since $C \leq C_L + C_S$, feasibility of this strategy is always guaranteed.

the smallest fractions $\phi_L$ and $\phi_S$, respectively. Although greedy, HPA is highly efficient and enables fast post-hoc compression of $\{\widehat{X}_i\}_{i=1}^N$ across a wide range of parameter budgets, without solving the intractable problem in (8). Please refer to Appendix D for a visualization of the HPA strategy.

# 5. Experiments

In this section, we evaluate the effectiveness of SALAAD for pretraining LLMs. All experiments are conducted on NVIDIA A100 or H100 GPUs. Code is available at https://github.com/HaoMAFRLu/salaad.

## 5.1. Pretraining

**Experimental setup.** We follow the experimental settings in Han et al. (2024); Li et al. (2025) and pretrain our models on the C4 dataset (Raffel et al., 2020). We adopt LLaMA-based LLMs (Touvron et al., 2023), employing prenormalization with RMSNorm (Zhang & Sennrich, 2019) and SwiGLU activations (Shazeer, 2020) throughout the network. Training is performed without data repetition, and model sizes range from 60M to 1B parameters.

**Baseline.** We compare our method with a set of state-of-the-art baselines for parameter-efficient pretraining. Full-rank pretraining serves as the vanilla baseline, where all model parameters are optimized in full-rank form. We include LoRA (Hu et al., 2021) as a representative low-rank adaptation method. We further compare against recent approaches, including ReLoRA (Lialin et al., 2023), GaLore (Zhao et al., 2024), LORO (Mo et al., 2024), CoLA (Liu et al., 2025), SLTrain (Han et al., 2024), and LOST (Li et al., 2025). All methods are trained using the same number of training tokens to ensure a fair comparison.

**Hyperparameters.** For all experiments in this work, we use a unified hyperparameter configuration across model scales, demonstrating the scalability of our method. Specifically, we set the target effective rank ratio $\widehat{\Gamma}_L^\gamma = 0.15$ and target density $\widehat{\Upsilon}_S = 0.05$ for all blocks, including the embedding layer. The I-controller step sizes are set at different orders of magnitude, with $\Delta\alpha$ on the order of $10^{-1}$ and $\Delta\beta$ on the order of $10^{-3}$, while the block-wise penalty coefficient $\rho$ follows the empirical scaling law in (7). We use Adam as the base optimizer with zero weight decay.

**Results and practical considerations.** Table 1 compares SALAAD with representative baselines across model scales and reports three model variants produced at different stages of the SALAAD pipeline. Specifically, $X$ denotes the original trained weights without guaranteed SLR structure, $L+S$ is the structured surrogate model that is SLR by construction, and $\widetilde{L} + \widetilde{S}$ denotes the HPA-compressed model obtained under a fixed parameter budget.

Table 1 establishes the central empirical conclusion of this work: across model scales, SALAAD achieves state-of-the-art compression performance while preserving the original model architecture. Under comparable or smaller parameter budgets, both the structured surrogate model $L + S$ and its HPA-compressed variants $\widetilde{L} + \widetilde{S}$ consistently match or outperform existing methods in terms of perplexity. Moreover, the consistent improvements across model sizes indicate that the proposed block-wise adaptive regulation of rank and sparsity scales favorably. Under comparable parameter budgets, SALAAD achieves lower perplexity than SLTrain (Han et al., 2024), and while it slightly underperforms LOST (Li et al., 2025) in some settings, LOST introduces additional nonlinear layers that alter the underlying architecture. In contrast, SALAAD attains competitive performance without architectural modification, making it a plug-and-play framework that is well suited for architecture-preserving and deployment-compatible scenarios.

The experimental settings underlying this comparison reflect different optimization priorities. Since our primary objective is elastic deployment at inference time, we assume sufficient computational resources during training. Accordingly, to ensure stable SLR induction and isolate the core optimization mechanism, all SALAAD models in Table 1 are trained in float32, with inference conducted in bfloat16. Under this regime, we do not report training-time memory cost, prioritizing numerical stability over training efficiency, which is an important consideration given the two-stage optimization and proximal updates used.

By contrast, all baseline methods are trained in bfloat16, reflecting a design focus on training efficiency. For completeness, we additionally report results where SALAAD is trained entirely in bfloat16 in Appendix E. Although performance is moderately degraded relative to the float32 setting, SALAAD remains competitive with all baselines, indicating robustness to reduced numerical precision while being primarily designed for deployment-oriented scenarios.

To quantify the training-time overhead introduced by the two-stage optimization, Figure 2 reports a wall-clock breakdown for the 350M and 1B models on 8 NVIDIA H100 80 GB GPUs. The additional cost mainly comes from the ADMM updates, while inter-GPU synchronization and saving auxiliary variables account for a smaller fraction of the total runtime. Since the surrogate blocks are decoupled and can be distributed across devices, this overhead decreases substantially as the number of available GPUs increases. This overhead is consistent with the training-rich, deployment-constrained setting considered in this work, where additional training computation is exchanged for elastic post-training deployment.

**Emergent SLR structure of embedding layers.** Previous work on structured pretraining (Mo et al., 2024; Liu et al.,

*Table 1.* Comparison of perplexity and parameter count between SALAAD and representative baselines across model scales. For SALAAD, $X$ denotes the original trained network, $L + S$ the structured surrogate model, and $\widetilde{L} + \widetilde{S}$ the model obtained via HPA strategy under a fixed parameter budget. All SALAAD results are trained in `float32`, with inference conducted in `bfloat16`. The perplexity results for all the baselines trained in `bfloat16` are taken from LOST (Li et al., 2025). Lower perplexity (PPL) is better.

| METHOD | 60M (1.1B) | | 130M (2.2B) | | 350M (6.4B) | | 1B (13.1B) | |
|---|---|---|---|---|---|---|---|---|
| | PPL | PRM(M) | PPL | PRM(M) | PPL | PRM(M) | PPL | PRM(M) |
| FULL-RANK | 34.06 | 58 | 24.36 | 134 | 18.80 | 368 | 15.56 | 1339 |
| LORA | 34.99 | 58 | 33.92 | 134 | 25.58 | 368 | 19.21 | 1339 |
| RELORA | 37.04 | 58 | 29.37 | 134 | 29.08 | 368 | 18.33 | 1339 |
| GALORE | 34.88 | 58 | 25.36 | 134 | 18.95 | 368 | 15.64 | 1339 |
| LORO | 33.87 | 43 | 24.78 | 94 | 19.66 | 185 | 15.53 | 609 |
| COLA | 34.10 | 43 | 25.61 | 94 | 19.75 | 185 | 15.76 | 609 |
| SLTRAIN | 34.15 | 44 | 26.04 | 97 | 19.42 | 194 | 16.14 | 646 |
| LOST | 32.25 | 43 | 24.05 | 94 | 18.95 | 185 | 15.02 | 609 |
| **SALAAD** | | | | | | | | |
| $X$ | 31.59 | 58 | 22.90 | 134 | 18.30 | 368 | 14.74 | 1339 |
| $L + S$ | **31.26** | 50 | **22.65** | 126 | **18.04** | 276 | **14.72** | 905 |
| $\widetilde{L} + \widetilde{S}$ | 31.97 ($\kappa = 0.7$) | 44 | 23.90 ($\kappa = 0.6$) | 97 | 19.70 ($\kappa = 0.6$) | 194 | 15.07 ($\kappa = 0.8$) | 646 |

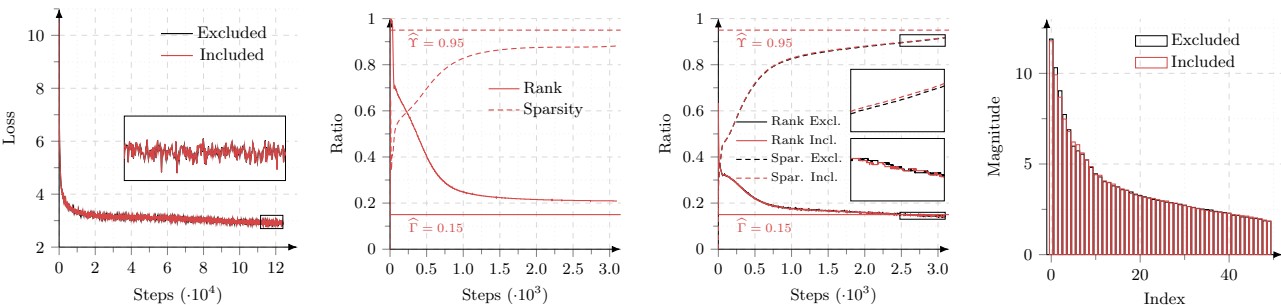

*(a)* Training loss with and without embedding layer    *(b)* Normalized convergence of rank ratio and density    *(c)* Normalized convergence of rank ratio and density    *(d)* Top singular values of the low rank component

*Figure 1.* Comparison of SALAAD training with and without embedding layer inclusion on a LLaMA-based 350M model. (a) Training loss trajectories. (b) Convergence of effective rank ratio and density in the embedding layer. (c) Convergence behavior of a randomly selected Transformer block. (d) Singular value spectra of the learned low-rank components. Overall, the results indicate that including the embedding layer does not affect training dynamics.

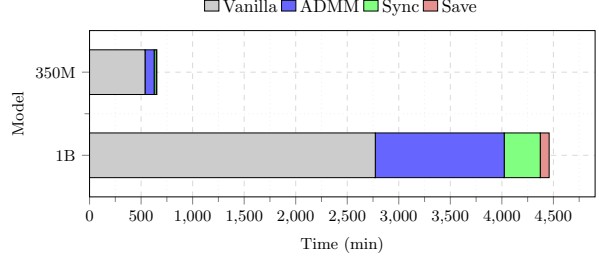

*Figure 2.* Training time breakdown for SALAAD on 8 `NVIDIA H100` 80 GB GPUs. All times are reported in minutes.

2025; Han et al., 2024; Li et al., 2025) consistently excludes the embedding layer and the LM head. Although the motivation is rarely stated explicitly, this design choice likely reflects either the perception that these layers do not exhibit inherent SLR structure, or the difficulty of heuristically spec-

ifying appropriate rank and sparsity levels for layers with atypical matrix dimensions, particularly in methods such as SLTrain (Han et al., 2024) and LOST (Li et al., 2025) that fix structural hyperparameters prior to training.

Benefiting from the adaptive design of our algorithm, we can seamlessly incorporate both the embedding layer and the LM head during training. Surprisingly, we find that the embedding layer naturally exhibits a benign SLR structure within our framework[4], characterized by smooth convergence of both rank and sparsity and negligible impact on the training dynamics of other blocks.

Figure 1 compares training a LLaMA-based 350M model with and without including the embedding layer under our

---

[4]The LM head does not exhibit this benign property; see Appendix H for additional evidence.

algorithm. For the embedding layer, we use exactly the same $(\Delta\alpha, \Delta\beta)$, target ratios $(\Gamma_L^\gamma, \Upsilon_S)$, and the same $\rho$ in (7) as for all other blocks, demonstrating the robustness of our adaptive scheme across different components. As shown in Figure 1a, the training losses under the two settings remain highly overlapping throughout training, even at the late stage. Figure 1b further shows that the effective rank ratio and sparsity in the embedding layer converge smoothly to low levels (approximately 21% rank ratio and 12% density). Finally, Figures 1c and 1d indicate that a randomly selected block exhibits nearly identical convergence behavior and singular value spectra[5] under both settings. Similar trends are consistently observed across other blocks and model sizes, additional results are provided in Appendix G.

**Flexible deployment with SALAAD.** One key advantage of SALAAD is its support for continuous elastic deployment across a wide range of parameter budgets without retraining. Figure 3 plots perplexity against parameter count for SALAAD models adapted to different budgets using the proposed HPA strategy, alongside vanilla models. For vanilla models, which are full-rank by construction, we first apply RPCA (Candès et al., 2011; Lin et al., 2010) to obtain an SLR approximation and then apply the same HPA procedure.

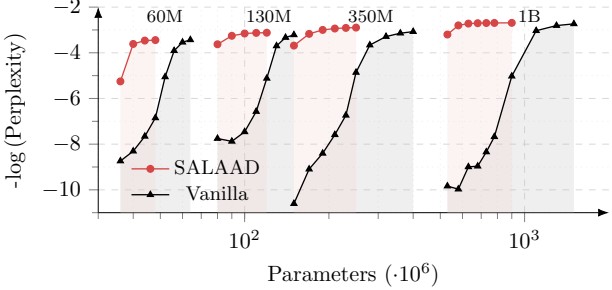

*Figure 3.* Perplexity versus parameter count for SALAAD models under different parameter budgets, compared with vanilla models. All results are obtained by applying HPA strategy.

As shown in Figure 3, SALAAD consistently outperforms vanilla models in perplexity across a wide range of parameter budgets. More importantly, the performance-capacity curves reveal a qualitative difference in deployment behavior: SALAAD enables smooth and continuous capacity scaling in weight space, whereas vanilla models exhibit less stable degradation as the parameter budget decreases. This suggests that elastic deployment depends critically on training-time induction of SLR structure, rather than post-hoc compression alone. Consequently, a single SALAAD checkpoint can be reliably adapted to diverse resource constraints without retraining or architectural modification, while the performance of compressed vanilla models degrades rapidly and limits their suitability for elastic deployment.

---

[5]Only the 50 largest singular values are displayed for clarity.

## 5.2. Downstream Evaluation

To complement perplexity-based evaluation, we further evaluate the LLaMA-based 1B model on standard downstream benchmarks using `lm-evaluation-harness` (Gao et al., 2021). All downstream evaluations are conducted in the zero-shot setting with the default task configurations. The benchmark suite covers multitask knowledge (MMLU), science question answering (ARC-C), causal commonsense reasoning (COPA), commonsense sentence completion (HellaSwag), yes/no reading comprehension (BoolQ), and physical commonsense reasoning (PIQA). Since these tasks are formulated as classification or multiple-choice evaluations in the harness, we report zero-shot accuracy, where higher values are better.

*Table 2.* Zero-shot downstream accuracy (%) of the LLaMA-based 1B models evaluated with default `lm-evaluation-harness` settings. $X$ denotes the SALAAD-trained dense model, and $\widetilde{L} + \widetilde{S}$ denotes its HPA-compressed model with 646M parameters.

| MODEL | MMLU | ARC-C | COPA | HELLASWAG | BOOLQ | PIQA |
|---|---|---|---|---|---|---|
| $X$ | 23.0 | 22.0 | 71.0 | 36.0 | 54.0 | 68.0 |
| $\widetilde{L} + \widetilde{S}$ | 23.0 | 22.0 | 69.0 | 36.0 | 52.0 | 69.0 |
| VANILLA | 23.0 | 22.0 | 69.0 | 34.0 | 55.0 | 69.0 |

As shown in Table 2, both SALAAD variants achieve downstream performance comparable to the vanilla 1B model across all evaluated benchmarks. In particular, the compressed companion model $\widetilde{L} + \widetilde{S}$ preserves the performance of $X$ on MMLU, ARC-C, and HellaSwag, while showing only small variations on COPA, BoolQ, and PIQA. These results indicate that the SLR structure induced by SALAAD does not lead to abnormal degradation or collapse on representative zero-shot downstream tasks, and that the perplexity improvements observed above are consistent with stable downstream behavior.

## 5.3. Ablation Studies

In this section, we conduct ablation studies to analyze the impact of key hyperparameters in our method.

**Effect of allocation ratio.** In the HPA strategy, the allocation ratio $\kappa$ controls how the parameter reduction budget is split between $\{L_i\}_{i=1}^N$ and $\{S_i\}_{i=1}^N$. To study its effect, we evaluate SALAAD models at different scales under multiple parameter budgets, sweeping $\kappa$ for each setting, as shown in Figure 4. Across all models, the optimal allocation ratio $\kappa^\star$ consistently lies within a narrow range (gray region), largely independent of the target budget. Performance degrades as $\kappa$ moves away from this region, and assigning a larger fraction of the budget to the low-rank component ($\kappa > 0.5$) yields better performance across all settings.

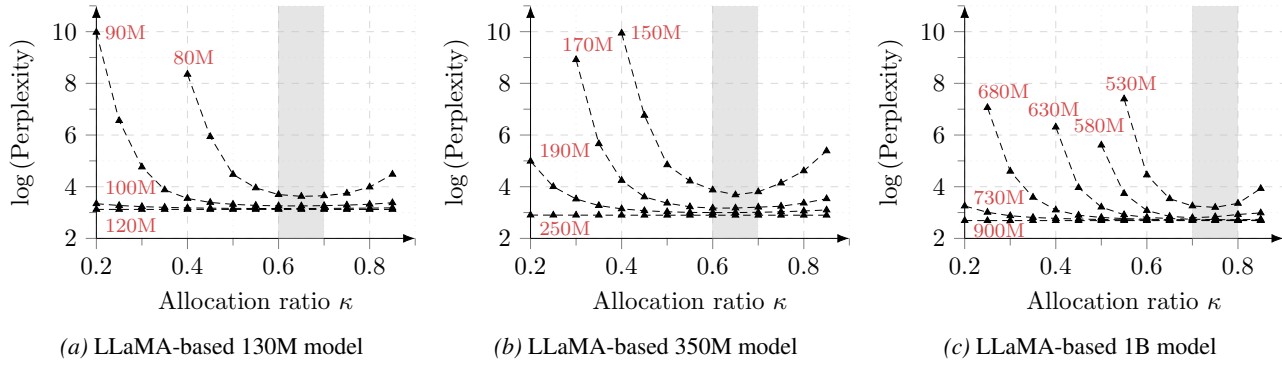

*(a) LLaMA-based 130M model*     *(b) LLaMA-based 350M model*     *(c) LLaMA-based 1B model*

*Figure 4.* Effect of allocation ratio $\kappa$ on model performance under different parameter budgets for LLaMA-based (a) 130M model, (b) 350M model, and (c) 1B model. The gray region indicates the relatively stable range of optimal allocation ratio $\kappa^{\star}$ across different budgets.

*Table 3.* Ablation study for the 350M model with $\rho = 5 \times 10^{-8}$.

| | | | | | |
|---|---|---|---|---|---|
| ABLATION ON $\Delta\beta$ ($\Delta\alpha = 0.2$) | | | | | |
| $\Delta\beta$ | 0.003 | 0.005 | 0.01 | 0.05 | 0.1 |
| PPL ($X$) | 18.62 | 18.98 | 19.42 | 19.90 | 19.97 |
| PPL ($L+S$) | 18.43 | 18.96 | 19.50 | 20.57 | 20.81 |
| PRM (M) | 241 | 228 | 221 | 218 | 218 |
| ABLATION ON $\Delta\alpha$ ($\Delta\beta = 0.005$) | | | | | |
| $\Delta\alpha$ | 0.08 | 0.1 | 0.15 | 0.18 | 0.2 |
| PPL ($X$) | 18.25 | 18.45 | 18.79 | 18.92 | 18.97 |
| PPL ($L+S$) | 17.97 | 18.23 | 18.63 | 19.07 | 18.96 |
| PRM (M) | 268 | 257 | 239 | 234 | 228 |

*Table 4.* Ablation on $\rho$ under different fixed $(\Delta\alpha, \Delta\beta)$ pairs.

| $\rho$ | PPL ($X$) | PPL ($L+S$) | PRM (M) |
|---|---|---|---|
| $\Delta\alpha = 0.1,\ \Delta\beta = 0.01$ | | | |
| $5\times10^{-8}$ | 18.45 | 18.23 | 257 |
| $1\times10^{-7}$ | 19.53 | 19.26 | 221 |
| $\Delta\alpha = 0.1,\ \Delta\beta = 0.05$ | | | |
| $5\times10^{-8}$ | 19.02 | 18.92 | 246 |
| $1\times10^{-7}$ | 20.23 | 20.18 | 215 |

**Sensitivity and robustness analysis.** We analyze the sensitivity of SALAAD to its key hyperparameters, including the step sizes $(\Delta\alpha, \Delta\beta)$, and the block-wise penalty coefficient $\rho$. All ablation studies are conducted on the LLaMA-based 350M model, with results summarized in Tables 3 and 4.

As $\Delta\alpha$, $\Delta\beta$, or $\rho$ increases, model perplexity generally rises while compressibility improves, reflecting more aggressive structural updates. This trade-off is expected, as larger step sizes or stronger structural regularization accelerate the adjustment of rank and sparsity at the cost of performance. Notably, increasing $\rho$ has an effect analogous to enlarging the step sizes, and thus fixing $(\Delta\alpha, \Delta\beta)$ while tuning $\rho$ provides an effective and low-dimensional strategy for balancing performance and compressibility. Additional results are reported in Appendix I.

## 6. Conclusion

We presented SALAAD, a plug-and-play framework for inducing SLR structure in LLMs during pretraining. By jointly optimizing dense weights and structured surrogates, SALAAD enables stable SLR induction without architectural modification. A single SALAAD training yields a structured model that supports flexible adaptation to diverse parameter budgets via the proposed HPA strategy during deployment. Across multiple model scales, SALAAD achieves competitive perplexity while offering greater deployment flexibility than existing approaches. Finally, we observe that embedding layers naturally exhibit benign SLR behavior under SALAAD, highlighting the broad applicability of the framework.

## Acknowledgements

Hao Ma gratefully acknowledges support from the Center for Learning Systems. Michael Muehlebach thanks the German Research Foundation for the support.

## Impact Statement

This paper presents work whose goal is to advance the field of machine learning by improving the efficiency and flexibility of large language model deployment. The techniques studied in this work focus on model compression and structured optimization, which may help reduce compute and memory costs in practical systems. We do not foresee any direct negative societal consequences arising specifically from this work.

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

# A. Limitations of Post-hoc Sparse and Low-Rank Decomposition

In this section, we examine the limitations of post-hoc SLR decomposition via RPCA. We show that, without SLR-aware training, weight matrices learned by standard optimization do not admit sufficiently structured SLR decompositions, rendering post-hoc truncation ineffective.

**Post-hoc RPCA on standard-trained weights.** We apply the RPCA method (Candès et al., 2011; Lin et al., 2010) to the weight matrices of a well-trained LLaMA 1B model trained with standard optimization, i.e., without any SLR-aware regularization. In addition, we include a LLaMA 3.2 3B model as a supplementary experiment to further evaluate the generality of our observations. For each Transformer block, we decompose the self-attention projections (Q/K/V/O) and MLP projections (Gate/Up/Down) into SLR components using RPCA, and report the recovered effective rank ratios and sparsity levels in Figure 5 for representative shallow, middle, and deep layers.

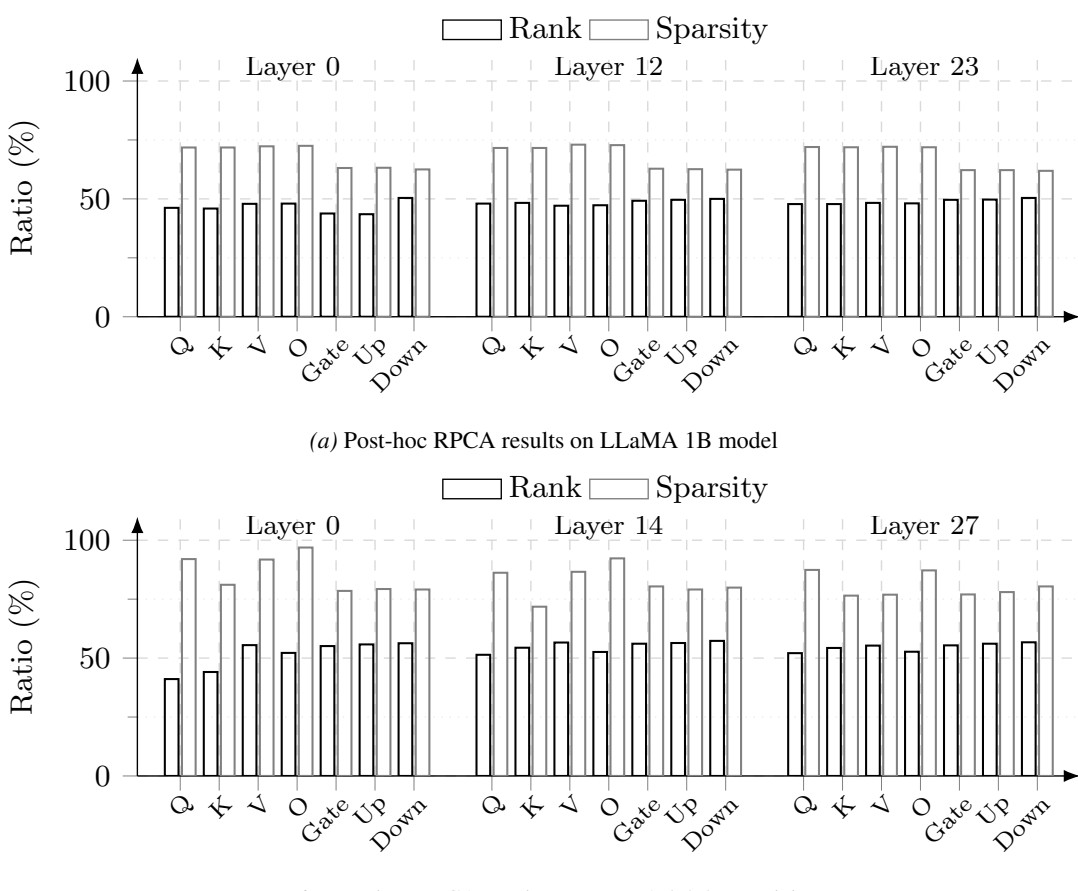

*(a)* Post-hoc RPCA results on LLaMA 1B model

*(b)* Post-hoc RPCA results on LLaMA 3.2 3B model

*Figure 5.* Post-hoc RPCA results on standard-trained models. (a) Results for the LLaMA 1B model. (b) Results for the LLaMA 3.2 3B model. For each model, representative layers from shallow, middle, and deep regions are selected, and for each layer the effective rank ratios and sparsity levels obtained after RPCA decomposition are reported for different matrix types.

Across both the 1B (see Figure 5a) and 3B (see Figure 5b) models, post-hoc RPCA yields decompositions with consistently weak SLR characteristics. The recovered low-rank components exhibit high effective rank ratios, while the sparse components show only moderate sparsity, with similar behavior observed across layers and matrix types. Quantitatively, the 1B model has an average effective rank ratio of $48.4\% \pm 2.0\%$ and an average sparsity level of $68.1\% \pm 5.0\%$, while the 3B model shows $54.6\% \pm 2.2\%$ and $81.6\% \pm 6.9\%$, respectively. These statistics remain far from the strongly low-rank or highly sparse regime required for effective structured compression. Taken together, these results indicate that, in the absence of SLR-aware training, standard optimization does not drive weight matrices toward the SLR manifold. Consequently, post-hoc decomposition methods such as RPCA are insufficient to extract SLR structures that enable meaningful reductions in model size or memory cost.

**Sanity check: RPCA on SALAAD-trained weights.** As a sanity check, Figure 6 applies the same RPCA procedure to weights from a 1B model trained with SALAAD, where SLR structure is explicitly induced during training. For representative shallow, middle, and deep layers, the recovered effective rank ratios and sparsity levels are compared against the ground-truth statistics from the original SLR components. Specifically, we construct dense weights as $\widehat{X} = L + S$ and apply RPCA to recover the underlying decomposition. The results show that RPCA can approximately recover the

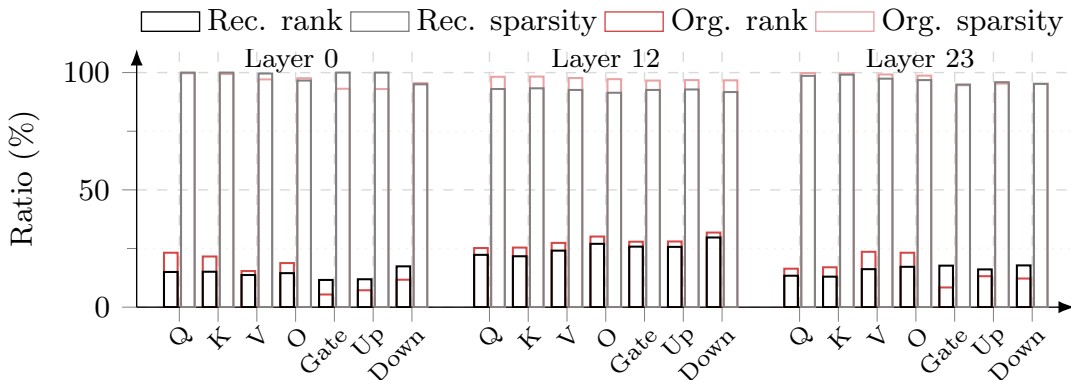

*Figure 6.* Post-hoc RPCA results on SALAAD-trained LLaMA-based 1B model. For several representative layers from shallow, middle, and deep regions, the effective rank ratios and sparsity levels obtained after RPCA decomposition are compared with the ground-truth values from the original SLR components learned by SALAAD. Black and gray boxes denote the recovered effective rank ratios and sparsity levels, respectively, while red and light-red boxes denote the corresponding ground-truth values.

latent SLR structure. Quantitatively, the ground-truth matrices exhibit an average effective rank ratio of $23.1\% \pm 8.4\%$ and sparsity of $94.2\% \pm 3.1\%$, while the recovered matrices achieve $25.3\% \pm 8.3\%$ and $97.4\% \pm 1.5\%$, respectively. Although not exact, the recovered statistics are close in magnitude, confirming that RPCA is capable of identifying SLR structure when such structure is genuinely present.

## B. Visual Overview of SALAAD Pipeline

Figure 7 illustrates the pretraining stage of SALAAD. Each selected block maintains a dense weight $X$ and surrogate variables $L$, $S$, and $Y$. This stage contains the ADMM-based sparse and low-rank adaptation step and the block-wise I-controller. The adaptation step updates the surrogate variables through proximal operations, while the I-controller adjusts the thresholds according to the discrepancy between the observed and target rank/sparsity levels. The resulting structured surrogates are then coupled back to the dense weights through the penalty loss.

Figure 8 illustrates the deployment stage of SALAAD. At deployment time, SALAAD deploys the learned structured surrogate $L + S$ rather than the dense training weights. Given a deployment budget $C$, HPA truncates the learned low-rank and sparse components by splitting the removable parameter budget according to $\kappa$ and deriving global scaling ratios $\phi_L$ and $\phi_S$. These ratios are applied proportionally to the learned SLR components in every block, producing compressed surrogates without retraining or architectural modification.

## C. Memory and Computational Cost Analysis

Algorithm 1 introduces additional memory usage, as each block stores three surrogate components $L$, $S$, and $Y$ in addition to the original weight matrices. In the large-scale LLM training regime, these additional costs remain manageable in practice.

**Memory Cost.** SALAAD targets elastic deployment on resource-constrained edge devices, whereas training is typically carried out on large GPU clusters. Accordingly, we assume that multiple GPUs are available during training. Additionally, SALAAD applies a block-wise I-controller that regulates rank and sparsity independently for each block. As a result, all surrogate blocks $\{L_i, S_i, Y_i\}_{i=1}^N$ are fully decoupled and can be updated in parallel. If the model contains $N$ blocks, these can be distributed across multiple GPUs so that each GPU handles at most $\lceil N/P \rceil$ surrogate blocks, where $P$ is the number of GPUs.

To illustrate the scale, consider a 70B-class Transformer such as LLaMA-70B, which contains 560 projection matrices

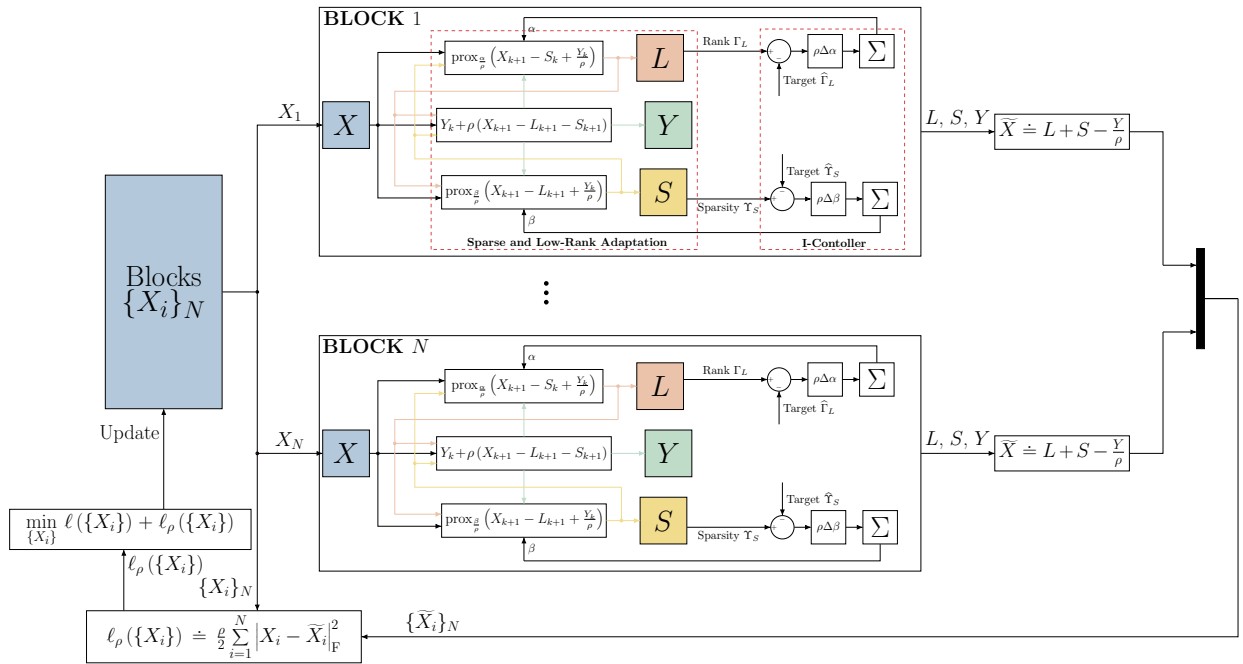

*Figure 7.* Pretraining stage of SALAAD. The dense weights are updated by the task loss and SLR-inducing penalty, while each selected block performs ADMM-style sparse and low-rank adaptation followed by I-controller updates for the rank and sparsity thresholds.

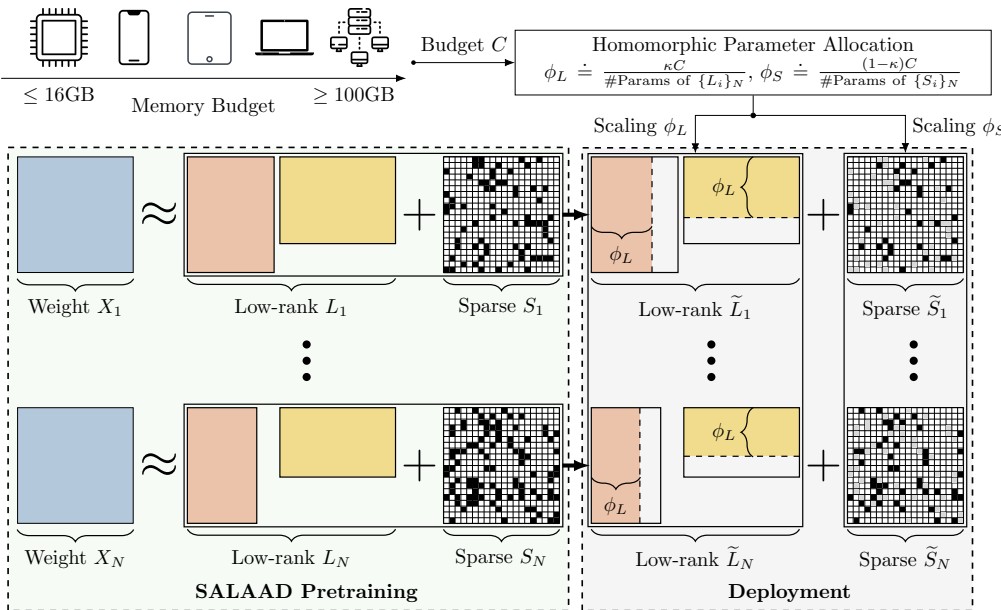

*Figure 8.* Deployment stage of SALAAD. SALAAD deploys the learned structured surrogate $L + S$ and uses HPA to map a memory budget to global scaling ratios $\phi_L$ and $\phi_S$, which truncate the learned low-rank and sparse components across blocks.

(blocks). When training on $P \geq 560$ GPUs, a configuration commonly used in large-model industrial training, each GPU is responsible for at most one block, resulting in an additional per-GPU memory cost of roughly 0.4-1.0 GB depending on the block type. In such large-scale and resource-intensive settings, this overhead constitutes a constant increase in per-GPU memory usage and may require a modest reduction in batch size.

This additional memory overhead is accompanied by the structural benefits provided by SALAAD, including improved perplexity during training and a structured surrogate model that supports elastic deployment on low-resource devices.

Moreover, in non-extreme settings, such as models below tens of billions of parameters or training on larger GPU clusters, the number of surrogate blocks per GPU decreases proportionally, reducing the per-GPU overhead to a small fraction of the available memory on modern accelerators (e.g., 40-80 GB on A100 or H100). In such regimes, the required memory concession becomes substantially smaller and is often negligible in practice.

**Computational Cost.** The dominant source of additional computation in Algorithm 1 arises from the SVD operations performed during the second-stage optimization. As discussed above, when surrogate blocks are evenly distributed across $P$ GPUs, where $P \geq N$, each GPU is responsible for at most one block, so the average per-GPU overhead per epoch can be expressed as $\epsilon J/K$, where $\epsilon$ denotes the cost of a single SVD, $J$ is the number of second-stage iterations, and $K$ is the number of first-stage iterations. For fixed $\epsilon$, the overhead scales linearly with $J$ and inversely with $K$.

In practice, we set $J = 1$ and find this choice to be both efficient and effective. There are several reasons for this design. First, during the early stages of training, the model weights $\{X_i\}_{i=1}^N$ do not exhibit stable SLR structures unless specifically initialized to do so. Consequently, repeatedly applying second-stage updates cannot reliably recover a meaningful SLR decomposition from $X$ in this phase, and a single update is sufficient. Second, with a properly chosen $\rho$, the algorithm guarantees that $|X - \widehat{X}|_\mathrm{F}$ remains bounded throughout training (see the discussion in Appendix F). Since our goal is not to recover an exact SLR decomposition at every step, but rather to induce a structurally regularized surrogate that co-evolves with $X$, one second-stage update is adequate. Empirically, the reconstruction error decreases naturally as training progresses. Third, using $J = 1$ yields a more stable interaction between the first-stage optimization (which updates $X$ using stochastic gradients) and the second-stage optimization (which projects towards the SLR manifold). Larger values of $J$ tend to over-regularize early in training and may interfere with the gradients, whereas a single second-stage step provides a gentle and stable form of structural correction.

For a block of size $8192 \times 8192$, the full SVD implemented by `torch.linalg.svd` costs on the order of $6.6 \times 10^{12}$ FLOPs, and for a block of size $8192 \times 22016$, about $1.0 \times 10^{13}$ FLOPs. With $J = 1$, the second-stage updates every $K = 40$ first-stage iterations and at most one block assigned per GPU. This corresponds to an average SVD overhead of approximately $1.6 \times 10^{11}$-$2.6 \times 10^{11}$ FLOPs per GPU per training iteration (i.e., roughly 0.16-0.26 TFLOPs), which is small compared to the $10^{13}$-$10^{14}$ FLOPs required for the forward-backward pass of a 70B-class model.

## D. Visualization of Homomorphic Parameter Allocation

Figure 9 provides a visualization of how the HPA strategy operates across Transformer blocks. The core idea of HPA is to

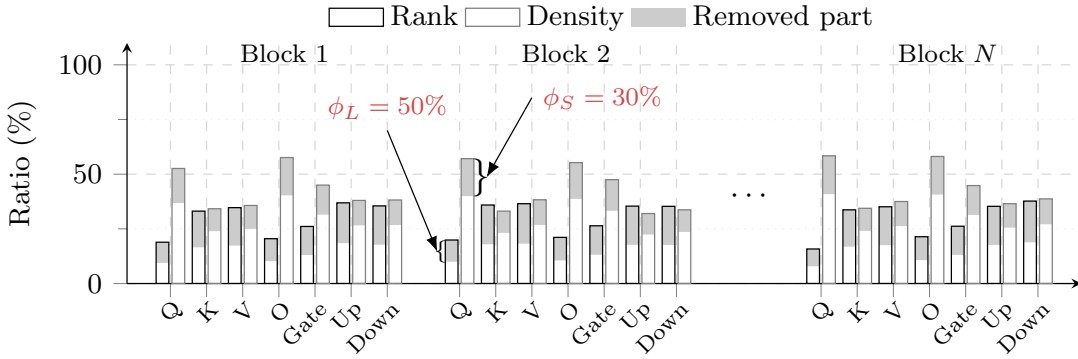

*Figure 9.* Visualization of Homomorphic Parameter Allocation (HPA) across Transformer blocks. Each block contains a low-rank component $L$ and a sparse component $S$, with varying ranks and sparsity levels. HPA enforces uniform proportional reductions $\phi_L$ and $\phi_S$ across all blocks, maintaining structural homomorphism.

enforce a global parameter budget $C$ together with a prescribed allocation ratio $\kappa$, from which two global scaling factors, $\phi_L$ and $\phi_S$, are derived according to (9) to control the reduction of the low-rank components $\{L_i\}_{i=1}^N$ and the sparse component $\{S_i\}_{i=1}^N$, respectively. Once these global ratios are determined, HPA applies the same proportional reduction to every block, rather than introducing block-specific thresholds or heuristics.

In the figure, this mechanism is illustrated using $\phi_L = 50\%$ and $\phi_S = 30\%$. For each Transformer block, the fraction of parameters removed from the low-rank component $L$ is fixed at $50\%$ of its original size, while the fraction removed from the

sparse component $S$ is fixed at $30\%$. Although different blocks exhibit substantially different absolute ranks and sparsity levels, the relative reduction ratios remain identical across blocks, thereby enforcing structural homomorphism throughout the network.

This visualization highlights the global-ratio, local-execution nature of HPA. By enforcing uniform proportional reductions across all blocks, HPA enables coordinated and continuous capacity control under a global budget constraint, without relying on block-wise sensitivity estimation or heuristic pruning rules.

## E. Training with `bfloat16`

In this section, we present additional training results of SALAAD with `bfloat16` for both training and inference as shown in Table 5. In practice, we observe that stable training under `bfloat16` can be achieved by slightly increasing

*Table 5.* Training results of SALAAD with the embedding layer included for the 60M, 130M, and 350M models. All models are trained in `bfloat16`, and inference is also performed under `bfloat16`.

| | 60M (1.1B) | | 130M (2.2B) | | 350M (6.4B) | |
|---|---|---|---|---|---|---|
| METHOD | PPL | PRM(M) | PPL | PRM(M) | PPL | PRM(M) |
| $X$ | 32.87 | 58 | 24.26 | 134 | 19.03 | 368 |
| $L + S$ | 32.60 | 43 | 24.18 | 129 | 18.93 | 287 |
| $\widetilde{L} + \widetilde{S}$ | 32.60 | 43 | 25.67 ($\kappa = 0.8$) | 97 | 19.73 ($\kappa = 0.75$) | 194 |

the penalty coefficient $\rho$ compared to the `float32` setting, while keeping $\Delta\alpha$ and $\Delta\beta$ unchanged. From an optimization perspective, this adjustment can be interpreted as rebalancing the effective regularization strength under reduced numerical precision. Since $\rho$ directly controls the strength of the structural constraint in the augmented objective, a modest increase in $\rho$ helps maintain a comparable update scale in the corresponding proximal updates. As discussed in Section 5.3 and Appendix I, increasing $\rho$ under fixed $\Delta\alpha$ and $\Delta\beta$ may lead to a moderate degradation in the final perplexity. Nevertheless, across all `bfloat16` experiments reported in Table 5, SALAAD remains highly competitive relative to all baselines. In all `bfloat16` experiments reported in Table 5, we additionally include the embedding layer during training, which empirically leads to improved compression performance. Overall, these results indicate that SALAAD can be trained entirely in `bfloat16` with only minor and well-motivated adjustments to the training configuration.

## F. Learning Dynamics

In this section, we analyze the learning dynamics of SALAAD training across different model scales. Specifically, we examine the evolution of sparsity and effective rank ratio for a randomly selected block, together with the corresponding block-wise reconstruction error defined as

$$\delta_i \doteq |X_i - L_i - S_i|_{\mathrm{F}} ,$$

where $i$ denotes the block index. In addition, we report the training loss trajectory and the average reconstruction error over all blocks, given by

$$\bar{\delta} \doteq \frac{1}{N} \sum_{i=1}^{N} \delta_i = \frac{1}{N} \sum_{i=1}^{N} |X_i - L_i - S_i|_{\mathrm{F}} ,$$

where $N$ denotes the total number of blocks in the model.

Figure 10 presents the learning dynamics of SALAAD across model scales ranging from 60M to 1B parameters. Across all scales, the training loss decreases smoothly and converges throughout optimization (see Figures 10a, 10e, 10i, and 10m), indicating that the introduction of the structured surrogate model does not destabilize standard training dynamics. Meanwhile, both the average reconstruction error $\bar{\delta}$ across all blocks (see Figures 10b, 10f, 10j, and 10n) and the block-wise reconstruction error $\delta_i$ of a representative block (see Figures 10d, 10h, 10l, and 10p) remain well bounded over the entire training process, suggesting that the approximation relationship between the dense model and its SLR surrogate is effectively controlled. Importantly, the sparsity and effective rank ratio evolve adaptively during training rather than following a predefined schedule (see Figures 10c, 10g, 10k, and 10o), reflecting the ability of SALAAD to automatically regulate structural capacity

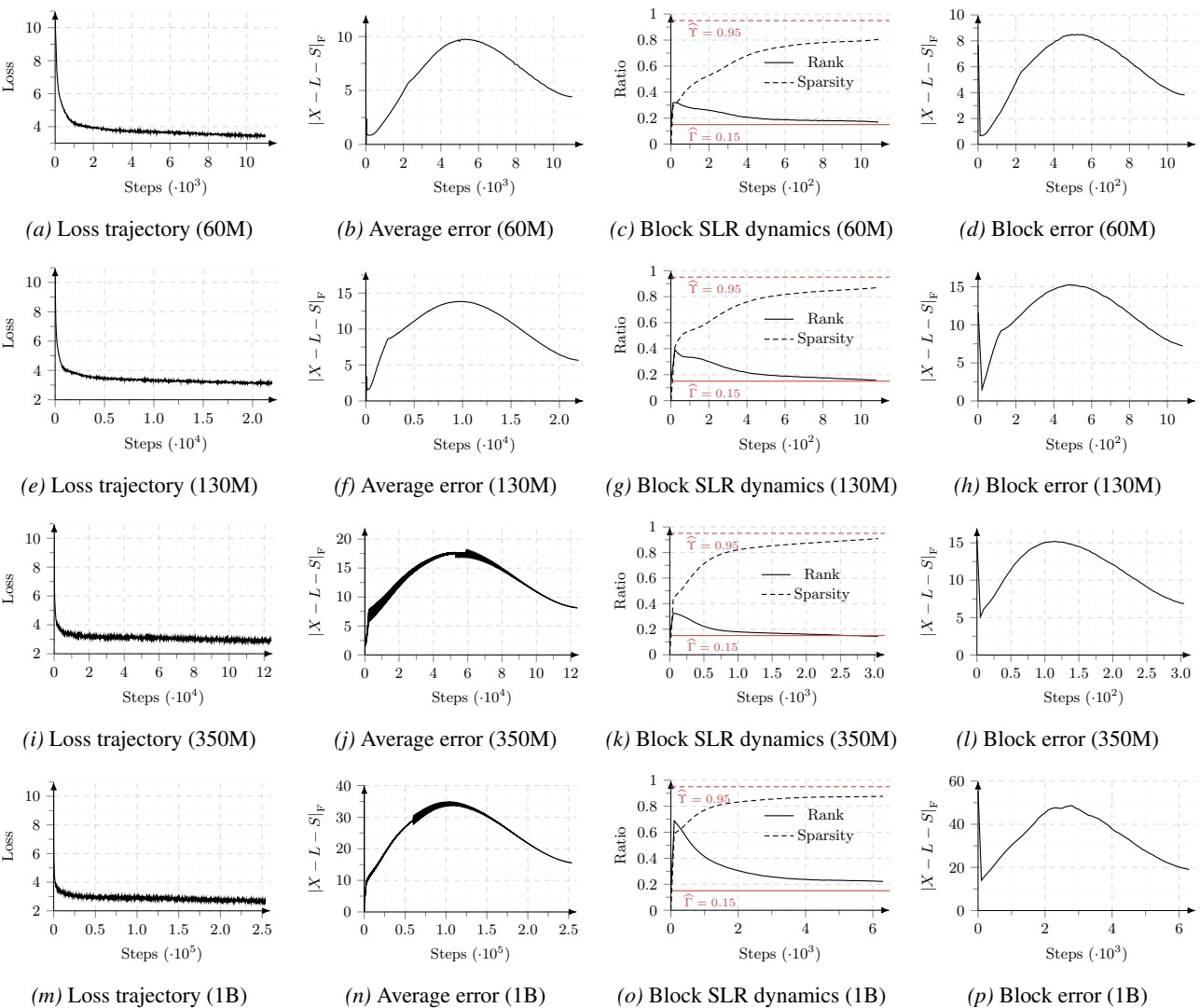

*Figure 10.* Learning dynamics of SALAAD training across model scales. Each row corresponds to a different model size, from top to bottom: 60M, 130M, 350M, and 1B. For each model scale, the four panels report the same set of training diagnostics: training loss (a, e, i, m), average reconstruction error across all blocks (b, f, j, n), sparsity and effective rank ratio evolution of a randomly selected block (c, g, k, o), and block-wise reconstruction error for a randomly selected block (d, h, l, p). Across all model scales, the training loss exhibits stable convergence, reconstruction errors remain bounded, and the sparsity and effective rank evolve adaptively, demonstrating self-adjusting capability of SALAAD in regulating SLR structure.

in response to the evolving optimization landscape. These behaviors are consistently observed across all model sizes, demonstrating that SALAAD exhibits stable and scalable training dynamics when applied to increasingly large models.

## G. Additional Analysis of Embedding Layer

In this section, we provide additional analysis on the effect of including the embedding layer in SALAAD training for the 60M and 130M models.

Figure 11 presents a comprehensive comparison of training dynamics with and without embedding layer inclusion across multiple aspects: (a, e) training loss trajectories, (b, f) embedding layer structural evolution, (c, g) representative Transformer block structural evolution, and (d, h) singular value spectra of learned low-rank components. The results indicate that including the embedding layer does not affect the training dynamics of SALAAD. The training loss trajectories with and without embedding layer inclusion remain highly overlapping throughout training, including the late stage (Figures 11a

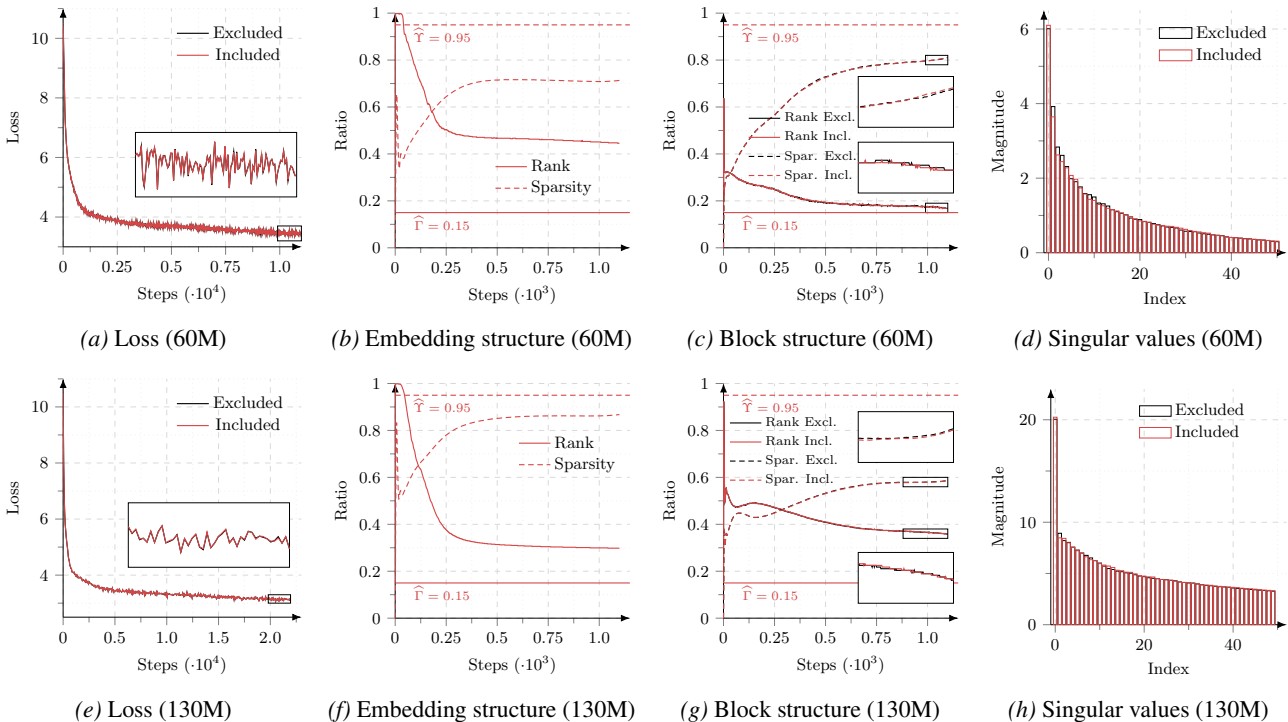

*(a)* Loss (60M)    *(b)* Embedding structure (60M)    *(c)* Block structure (60M)    *(d)* Singular values (60M)

*(e)* Loss (130M)    *(f)* Embedding structure (130M)    *(g)* Block structure (130M)    *(h)* Singular values (130M)

*Figure 11.* Effect of embedding layer inclusion in SALAAD training across model scales. The first row corresponds to the 60M model and the second row to the 130M model. (a, e) Training loss trajectories with and without embedding layer inclusion remain highly overlapping throughout training, including the late stage. (b, f) The embedding layer exhibits smooth and well-behaved convergence of effective rank and density. (c, g) A representative Transformer block shows highly consistent structural convergence under the two training settings. (d, h) The singular value spectra of the learned low-rank components are highly consistent.

and 11e). The embedding layer exhibits smooth and well-behaved convergence of effective rank and density (Figures 11b and 11f). A randomly selected Transformer block shows highly consistent structural convergence under the two training settings (Figures 11c and 11g). The singular value spectra of the learned low-rank components are highly consistent, indicating that including the embedding layer does not affect training dynamics (Figures 11d and 11h). Overall, these findings suggest that including the embedding layer in SALAAD training is a viable strategy to enhance compression without adversely impacting the training dynamics of the model.

Table 6 summarizes results for models with varying sizes when the embedding layer is included. Embedding layer inclusion introduces additional compression redundancy, resulting in nearly unchanged perplexity compared to Table 1, while further improving overall model compressibility. All models are trained in `float32`, with inference performed in `bfloat16`.

*Table 6.* Training results of SALAAD with the embedding layer included for the 60M, 130M, and 350M models. All models are trained in `float32`, and inference is performed using `bfloat16`.

| METHOD | 60M (1.1B) | | 130M (2.2B) | | 350M (6.4B) | |
|---|---|---|---|---|---|---|
| | PPL | PRM(M) | PPL | PRM(M) | PPL | PRM(M) |
| $X$ | 31.61 | 58 | 22.94 | 134 | 18.31 | 368 |
| $L + S$ | 31.29 | 48 | 22.48 | 113 | 18.02 | 231 |
| $\widetilde{L} + \widetilde{S}$ | 31.39 $(\kappa = 0.8)$ | **43** | 22.98 $(\kappa = 0.55)$ | **94** | 19.20 $(\kappa = 0.6)$ | **185** |

## H. Non-Benign SLR Behavior of the LM Head

The experimental results show that the LM head does not exhibit such benign SLR behavior. Figure 12 summarizes this

behavior on the LLaMA-based 60M model. As shown in Figure 12b, a smaller penalty coefficient $\rho$ fails to induce a stable

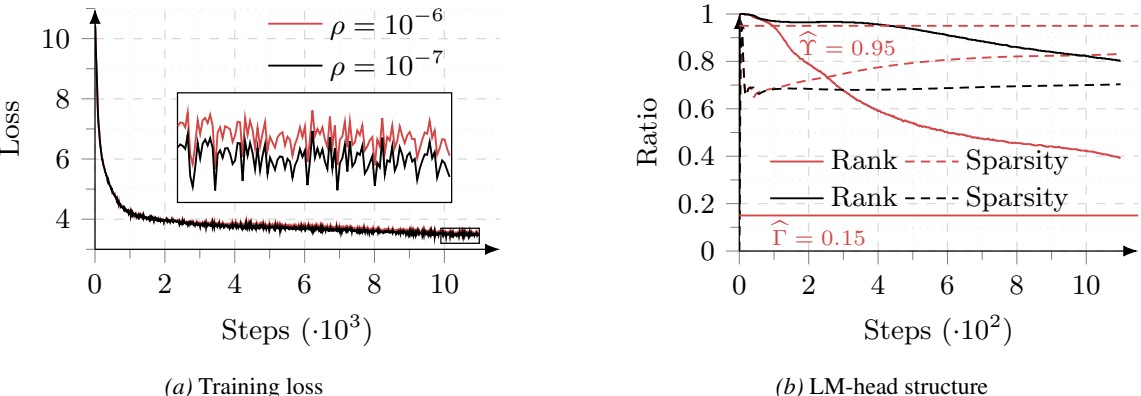

*(a)* Training loss           *(b)* LM-head structure

*Figure 12.* Behavior of the LM head under SLR-inducing regularization on the LLaMA-based 60M model. Red curves correspond to $\rho = 10^{-6}$ and black curves correspond to $\rho = 10^{-7}$. (a) Increasing $\rho$ degrades the training loss. (b) A smaller $\rho$ fails to induce a stable SLR structure in the LM head, whereas increasing $\rho$ strengthens the induced SLR structure.

SLR structure in the LM head. Increasing $\rho$ strengthens the induced SLR structure, but this comes at the cost of degraded training loss, as shown in Figure 12a. This behavior contrasts with the embedding layer, where a small $\rho$ is sufficient to induce a strong SLR structure and increasing $\rho$ does not noticeably affect the training loss. These results indicate that the LM head differs qualitatively from the embedding layer: it does not exhibit the same benign SLR property and is more likely to perturb the overall training dynamics when forced into an SLR form. We therefore exclude the LM head from the default SALAAD configuration and leave a more systematic investigation of its structural role to future work.

## I. Additional Ablation Study

### I.1. Effects of Structural Hyperparameters

We first present additional ablation studies on the 130M model trained with SALAAD. We investigate the effects of varying $\Delta\beta$, $\Delta\alpha$, and $\rho$ on model performance and parameter count. In Table 7, we vary $\Delta\beta$ while keeping $\rho = 1 \times 10^{-7}$ and $\Delta\alpha = 0.5$ fixed. The results indicate that smaller $\Delta\beta$ values lead to better PPL scores, albeit with a higher parameter count. A similar trend is observed in Table 8, where we vary $\Delta\alpha$ while fixing $\rho = 1 \times 10^{-7}$ and $\Delta\beta = 0.005$. Smaller $\Delta\alpha$ values also result in improved PPL scores but at the cost of increased parameters.

*Table 7.* Ablation on $\Delta\beta$ with $\rho = 1 \times 10^{-7}$ and $\Delta\alpha = 0.5$.

| $\Delta\beta$ | 0.0005 | 0.005 | 0.5 |
|---|---|---|---|
| PPL $(X)$ | 21.92 | 22.40 | 25.48 |
| PPL $(L+S)$ | 21.86 | 22.34 | 26.29 |
| PRM (M) | 138 | 122 | 99 |

*Table 8.* Ablation on $\Delta\alpha$ with $\rho = 1 \times 10^{-7}$ and $\Delta\beta = 0.005$.

| $\Delta\alpha$ | 0.005 | 0.05 | 0.2 |
|---|---|---|---|
| PPL $(X)$ | 21.72 | 22.30 | 23.22 |
| PPL $(L+S)$ | 21.77 | 22.14 | 22.97 |
| PRM (M) | 173 | 145 | 113 |

Table 9 explores the impact of varying $\rho$ across different $(\Delta\alpha, \Delta\beta)$ pairs. The results show that lower $\rho$ values generally yield better PPL scores. However, this also leads to a higher parameter count.

*Table 9.* Ablation on $\rho$ under different $(\Delta\alpha, \Delta\beta)$ pairs. Parameters are reported in millions.

| $\rho$ | $\Delta\alpha = 0.005$ | | | | | | | | | $\Delta\alpha = 0.05$ | | | | | | | | | $\Delta\alpha = 0.5$ | | | | | | | | |
|---|---|---|---|---|---|---|---|---|---|---|---|---|---|---|---|---|---|---|---|---|---|---|---|---|---|---|---|
| | 0.0005 | | | 0.005 | | | 0.05 | | | 0.0005 | | | 0.005 | | | 0.05 | | | 0.0005 | | | 0.005 | | | 0.05 | | |
| | X | L | PRM | X | L | PRM | X | L | PRM | X | L | PRM | X | L | PRM | X | L | PRM | X | L | PRM | X | L | PRM | X | L | PRM |
| $10^{-8}$ | 21.70 | 21.70 | 190 | 21.70 | 21.94 | 183 | 21.71 | 21.99 | 182 | 21.71 | 21.69 | 171 | 21.71 | 21.66 | 180 | 21.69 | 21.77 | 173 | 21.71 | 21.69 | 153 | 21.79 | 21.75 | 151 | 22.15 | 22.28 | 140 |
| $10^{-7}$ | 21.68 | 21.66 | 180 | 21.72 | 21.77 | 173 | 21.71 | 21.84 | 172 | 21.83 | 21.77 | 153 | 22.31 | 22.14 | 145 | 22.47 | 22.26 | 143 | 21.92 | 21.86 | 138 | 22.40 | 22.34 | 122 | 25.48 | 26.29 | 99 |
| $10^{-6}$ | 22.40 | 22.32 | 149 | 22.52 | 22.42 | 148 | 22.55 | 22.44 | 148 | 24.04 | 23.65 | 116 | 26.32 | 25.74 | 100 | 27.00 | 26.38 | 99 | 24.49 | 24.08 | 104 | 28.09 | 27.78 | 83 | 29.97 | 30.43 | 80 |

## I.2. Effect of ADMM Update Frequency

We further study the effect of the relative update frequency between gradient updates and ADMM structural updates. Since each ADMM phase uses one structural update in these experiments, $K/J$ is controlled by the number of gradient steps between two ADMM updates. We compare $K/J \in \{5, 10, 20\}$ on the LLaMA-based 1B model while keeping all other hyperparameters fixed. The horizontal axis in Figure 13 denotes the number of ADMM updates rather than the number of gradient steps.

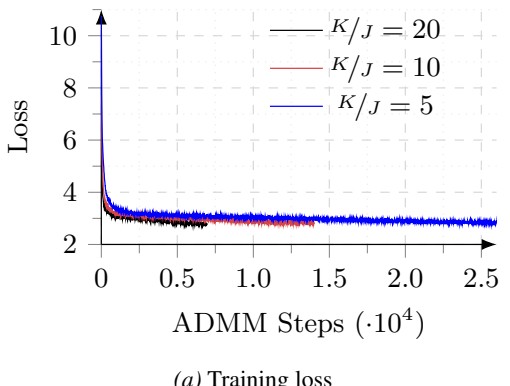
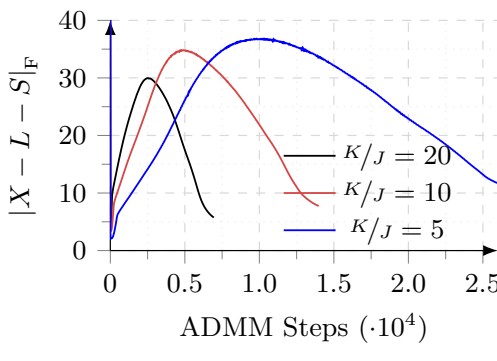

*(a)* Training loss          *(b)* Average reconstruction error

*Figure 13.* Effect of ADMM update frequency on the LLaMA-based 1B model. The horizontal axis denotes the number of ADMM updates. We compare $K/J = 5$, 10, and 20, corresponding to ADMM updates every 5, 10, and 20 gradient steps, respectively.

Across all three settings, the training loss converges stably, indicating that changing the ADMM update frequency does not destabilize optimization. The final training losses are 2.84, 2.84, and 2.78 for $K/J = 5$, 10, and 20, respectively. The average reconstruction error exhibits clearer dependence on the update schedule, with final values of 10.16, 7.74, and 5.73 for $K/J = 5$, 10, and 20. This suggests that $K/J$ changes how closely the structured surrogate tracks the dense weights during training, while the task loss remains comparatively robust across the tested range. In practice, we choose a moderate $K/J$ by balancing structural tracking quality and the computational cost of ADMM updates.

Table 10 further reports the final rank ratio and sparsity of randomly sampled selected blocks under the three update frequencies. Because all other hyperparameters are fixed, smaller $K/J$ corresponds to more frequent ADMM updates and therefore stronger SLR regularization. This trend is reflected consistently in Table 10: decreasing $K/J$ generally yields lower rank ratios and higher sparsity across the sampled blocks, indicating stronger structured compression. The stronger regularization also explains the slightly worse final training loss for $K/J = 5$ and 10 compared with $K/J = 20$, showing the expected trade-off between structural strength and task-loss optimization.

*Table 10.* Final rank ratio and sparsity for randomly selected blocks under different $K/J$ settings on the LLaMA-based 1B model. Rank ratio and sparsity are reported in percent.

| BLOCK | $K/J = 5$ | | $K/J = 10$ | | $K/J = 20$ | |
|---|---|---|---|---|---|---|
| | RANK RATIO | SPARSITY | RANK RATIO | SPARSITY | RANK RATIO | SPARSITY |
| EMBEDDING | 13.5 | 95.9 | 14.1 | 95.4 | 17.4 | 93.6 |
| LAYER 3.DOWN | 19.8 | 79.9 | 27.9 | 77.8 | 41.3 | 74.7 |
| LAYER 4.K | 14.7 | 92.9 | 16.5 | 90.4 | 22.5 | 85.4 |
| LAYER 8.Q | 16.0 | 86.3 | 22.6 | 81.1 | 30.8 | 77.7 |
| LAYER 11.GATE | 19.5 | 79.8 | 26.3 | 79.6 | 41.7 | 74.8 |
| LAYER 15.K | 18.0 | 82.0 | 25.2 | 78.5 | 32.7 | 75.6 |
| LAYER 18.K | 17.0 | 84.8 | 22.8 | 80.5 | 32.0 | 76.4 |
| LAYER 18.DOWN | 23.1 | 71.7 | 38.3 | 69.8 | 46.1 | 71.7 |
| LAYER 20.K | 15.1 | 88.8 | 20.8 | 82.9 | 29.6 | 78.0 |
| LAYER 23.GATE | 21.2 | 76.5 | 31.2 | 75.5 | 42.7 | 74.6 |

Overall, these ablation studies highlight the trade-offs between model performance and parameter efficiency when tuning the hyperparameters $\Delta\beta$, $\Delta\alpha$, $\rho$ and update frequency in the SALAAD framework.

