# OpenReview forum: "SALAAD: Sparse And Low-Rank Adaptation via ADMM for Large Language Model Inference"
_ICML.cc/2026/Conference — ICML 2026 regular_

### Official Review · Reviewer_C4Yo · 2026-02-26

**Soundness:** 3
**Presentation:** 3
**Significance:** 3
**Originality:** 3
**Overall Recommendation:** 4
**Confidence:** 4

**Summary:**

Low-Rank Approximation and Pruning are crucial techniques for large model compression. However, existing methods typically rely on heuristic designs, overlook the heterogeneity across layers and matrices, or require modifications to the specific model's architecture. Recognizing the importance of structured optimization during the training phase, SALAAD proposes a plug-and-play structured framework designed for large language model pre-training, utilizing ADMM as its theoretical core. Comparisons with multiple baselines demonstrate its effectiveness.

**Compliance With Llm Reviewing Policy:**

Affirmed.

**Final Justification:**

The author did not address all of my concerns, so I am maintaining my score.

**Key Questions For Authors:**

See Weaknesses & Questions.

**Limitations:**

yes

**Strengths And Weaknesses:**

Strengths:

- Proactively injects the SLR (Sparse and Low-Rank) structure into the optimization process as an intrinsic constraint.

- Breaks away from static and heuristic settings.

- Maintains high compatibility with the underlying network architecture.

Weaknesses & Questions:

- What is the inference efficiency (e.g., throughput) of the model after the training is completed?

- Can it support larger-scale models (e.g., 7B, 13B, or even 70B)? Model compression is arguably more meaningful on larger models.

- How effective is this method on different model architectures (e.g., Mixture of Experts / MoE)?

- Perplexity is not always a reliable metric. Could you demonstrate the performance differences between SALAAD and the baselines using more reliable downstream task accuracy metrics (such as GSM8K and MMLU)?

---

> ### Author Rebuttal · Authors · 2026-03-28
>
> Thank you for dedicating your time and expertise to review our submission. Please find our responses below.
> > Inference efficiency
>
> We believe that efficient deployment of large models on edge devices typically involves two relatively independent but equally important aspects: parameter compression, and computational efficiency and throughput during inference. This work primarily focuses on the former, namely inducing structures during training to enable effective compression at deployment. Accordingly, we use parameter budget as the primary evaluation metric. Moreover, in many edge-device scenarios, memory and storage themselves are key bottlenecks, making parameter compression an independently important objective. Given the scope of this work, theoretical and empirical analysis from the perspective of parameter compression is sufficient to support our main claims, without requiring additional throughput evaluation on specific hardware platforms.
>
> Furthermore, when the original matrix is approximated via SLR decomposition, especially under unstructured sparsity, inference efficiency typically does not improve and may even degrade. This is mainly due to irregular memory access patterns that negatively affect hardware execution efficiency, as also noted by the reviewers and prior works [1,2]. Therefore, parameter compression and inference acceleration do not necessarily align as optimization objectives in practice. In future work, we will introduce hardware-friendly structured sparsity (e.g., N:M or block sparsity [3,4]) to better balance compression effectiveness and execution efficiency. Importantly, our method does not restrict the incorporation of such structured constraints and is therefore compatible with existing acceleration techniques. Correspondingly, the mathematical formulation of each block can be extended as:
> $$
> \min_{X \in \mathcal{X}} \ell(X) + \alpha |L| + \sum_{i=1}^M \beta_i |S_i|_1
> $$
>
> $$
> \text{s.t. } X = L + S, \\\{S_i\\\}_{i=1}^M \text{ is a partition of } S
> $$
> where variables follow the same definitions as in (1) in the article.
>
> In summary, from the perspective of problem formulation, this work intentionally focuses on parameter compression rather than inference acceleration; the two are distinct but complementary directions.
> > Large-scale models
>
> Our method naturally scales to larger models, as its core idea is to induce structured representations during training, which does not depend on specific model scales or architectures. Empirically, we adopt a unified hyperparameter configuration across different scales and observe stable performance, demonstrating the robustness and scalability of our method.
>
> Due to computational constraints, this work focuses on evaluation on moderate-scale models, which is also common practice in related work. Moreover, for deployment on edge devices, 1B models are already non-trivial and represent a meaningful target for compression. But we agree that as model size increases, compression becomes even more important. In future work, we will extend our evaluation to larger models and assess performance in real-world deployment scenarios.
> > Different model architectures
>
> Our method is architecture-agnostic, as it operates on parameter matrices and does not rely on specific network designs, making it applicable to a wide range of architectures. For MoE models, it can be applied independently to each expert without affecting the routing mechanism, ensuring compatibility. While we focus on dense Transformers due to computational constraints, we expect the method to generalize to more complex architectures such as MoE and will explore this in future work.
> > Downstream tasks
>
> We report downstream evaluation results on three model variants: the model trained with our method ($X$), its compressed version ($\tilde{L}+\tilde{S}$, 646M), and a vanilla baseline model. The evaluation covers representative benchmarks in reasoning, commonsense, knowledge, and factual reliability. Under the zero-shot setting, all models exhibit the typical behavior of pretrained models without finetuning. Compared to the vanilla model, our method and its compressed variant achieve highly comparable performance, with only minor variations across individual benchmarks. No abnormal fluctuations or signs of collapse are observed, indicating stable and robust behavior.
> Model|MMLU|ARC-C|COPA|HellaSwag|BoolQ|PIQA|Winogrande
> -|-|-|-|-|-|-|-
> $X$|0.23|0.22|0.71|0.36|0.54|0.68|0.53
> $\tilde{L}+\tilde{S}$|0.23|0.22|0.69|0.36|0.52|0.69|0.53
> Vanilla|0.23|0.22|0.69|0.34|0.55|0.69|0.51
>
> Since the baselines do not release weights and full reproduction is infeasible under current time and computational constraints, we are unable to conduct fair comparisons under identical settings. We will include additional results and provide more comprehensive comparisons where possible in the revision.
>
> [1] Han et al., NeurIPS 2024 [2] Li et al., arXiv 2025 [3] Mishra et al., arXiv 2021 [4] Gray et al., arXiv 2017

---

> > ### Author Rebuttal · Reviewer_C4Yo · 2026-04-02
> >
> > Thanks for your reply.
> >
> > I believe that showcasing actual results and larger models would help highlight SALAAD's advantages even more effectively.

---

> > > ### Author Response · Authors · 2026-04-06
> > >
> > > Thank you for your helpful suggestion.
> > >
> > > We agree that evaluating SALAAD on larger-scale models would further strengthen the empirical evaluation and better highlight its advantages. Due to current computational resource constraints, we were not able to include such experiments at this stage. We will incorporate this in the revised version and evaluate the method on larger models as resources permit.

---

### Official Review · Reviewer_6Rsb · 2026-03-04

**Soundness:** 2
**Presentation:** 2
**Significance:** 3
**Originality:** 3
**Overall Recommendation:** 4
**Confidence:** 3

**Summary:**

The deployment of modern large language models under compute and memory constraints faces significant challenges. Existing compression methods based on sparse and low rank structures mostly rely on heuristic designs ignore network layer heterogeneity and even require modifications to the underlying model architecture. To address this the paper proposes SALAAD a plug and play framework designed to induce sparse and low rank structures for various model architectures during the training phase. This method builds a weight learning mechanism under an augmented Lagrangian framework and specifically introduces an adaptive integral controller to dynamically balance training loss and structural constraints. This controller can automatically adjust the penalty strength of rank and sparsity at the block level eliminate tedious manual rule settings enable each network layer to spontaneously adapt to its optimal physical form and precisely control the evolution of model capacity. Experiments on multi-scale models show that this method substantially reduces deployment memory consumption while maintaining excellent performance comparable to specialized compression methods.

**Compliance With Llm Reviewing Policy:**

Affirmed.

**Final Justification:**

I will maintain my current score of **4 Weak Accept**. This is a theoretically solid work. Although there were some issues, my concerns have been resolved. I thank the authors for their rebuttal. This paper will make a meaningful contribution to the community. I hope the authors can further improve the visual presentation of the charts and figures. This enhancement will make the reported data much more intuitive for readers to understand.

**Key Questions For Authors:**

The paper presents an innovative approach to large language model compression, but there are several significant issues that need to be addressed, particularly regarding logical consistency, experimental fairness, and practical evaluation. Below are the key concerns raised during the review process.

Major Question 1: Logical Inconsistency Between Training and Deployment Stages Regarding Layer Heterogeneity.
The design logic between the training and deployment stages in this paper contains a severe conflict. The authors criticize structural designs that ignore layer heterogeneity in the abstract and introduction. To address this an adaptive controller is introduced to achieve dynamic adjustment of rank and sparsity at the block level. However when introducing the homomorphic parameter allocation strategy the authors abandon this heterogeneity argument. The paper explicitly states that all network blocks adopt a shared globally uniform ratio for scaling and pruning. This globally consistent pruning method destroys the finely learned structures from the training stage and reverts to the layer agnostic paradigm criticized by the authors thereby obliterating the sensitivity differences between shallow and deep network layers. Please provide a theoretical justification for the representational capacity of this theoretical compromise in the main text. Furthermore it is strongly recommended to add baseline comparison experiments evaluating the perplexity of the current globally uniform scaling strategy against a non uniform pruning strategy based on layer sensitivity awareness to validate the rationality of the current design.

Major Question 2: Double Standards in Precision During Experimental Comparisons Lead to Unfair Evaluation.
The paper presents superior compression performance compared to baseline methods in its core tables and frames this as a primary contribution. However the authors explicitly admit in the main text that all SALAAD models are trained under float32 precision while all baseline methods used for comparison are trained under bfloat16 precision. The later sections reconfirm that SALAAD is trained in float32 precision. Comparing models trained with high precision against baselines trained with low precision severely violates the experimental principle of controlling variables. Higher numerical precision naturally improves optimization stability and yields lower perplexity. Although the appendix provides some bfloat16 results employing a double standard in the main text obscures the fragile dependence of the algorithm on numerical precision. It is recommended that the authors rerun the baseline comparison experiments under exactly identical numerical precision.

Major Question 3: Lack of Physical Performance Evaluation on Real Edge Hardware.
The core motivation of this paper is to solve the deployment challenges of large language models on resource constrained edge devices. The authors specifically mention physical hardware with limited memory such as Raspberry Pi 5 and NVIDIA Jetson in the text. Despite a strong hardware orientation the evaluation is limited to theoretical parameter counts and language model perplexity. The algorithm typically induces unstructured sparse matrices. Unstructured sparsity often struggles to achieve real acceleration on actual hardware due to irregular memory access patterns. The absence of actual latency or throughput tests on target hardware leaves the claims about elastic deployment without empirical support. The authors should supplement end to end inference speed and physical memory usage tests of the models on real edge devices.

Major Question 4: Extreme Sensitivity of Hyperparameters Contradicts the Plug and Play Framework Claims.
The paper repeatedly emphasizes that SALAAD is a plug and play pretraining framework. Such a description usually implies that the algorithm possesses good robustness and does not require extremely tedious tuning. However the ablation studies in the appendix reveal that the model is highly sensitive to the newly introduced global hyperparameters. Data in Table 6 show that modifying a single parameter from 0.005 to 0.5 causes the perplexity to drastically degrade from 22.40 to 25.48. Table 8 further demonstrates that minor perturbations in the penalty coefficient lead to severe fluctuations in model performance. This fragility requires developers to conduct expensive grid searches when facing new tasks just to make the model converge. The authors need to provide a set of parameter initialization guidelines that do not require extensive trial and error to fulfill their promises of practicality.

**Limitations:**

The paper lacks a comprehensive and transparent discussion of its inherent limitations in the main text. Several critical vulnerabilities that hinder its practical deployment are either downplayed or relegated to the appendix.
1. The limitation of dependence on numerical precision is downplayed.
The authors do not deeply explore the vulnerability of this algorithm to high numerical precision in the limitations section of the main text but instead hide the performance degradation results caused by half-precision floating-point training in Appendix D. This evades the critical flaw that the proposed method currently struggles to perfectly adapt to the mainstream low-precision training paradigms of modern large language models.
2. The limitation of hyperparameter sensitivity is circumvented.
Although the main text repeatedly advertises a plug-and-play nature the authors fail to discuss the practical difficulties of global parameter tuning. While they demonstrate in the ablation study of Appendix G that perturbations in the penalty coefficient lead to severe fluctuations in perplexity they do not provide a theoretical explanation or remedial guidelines for this fragility in the main text.

**Strengths And Weaknesses:**

This paper presents a theoretically innovative method based on the alternating direction method of multipliers for inducing sparse structures, with significant practical implications. However, issues such as unfair experimental design and lack of performance testing limit its practical application value.

Strengths：
1. Soundness. The mathematical derivation of introducing optimization algorithms to induce sparse and low rank structures during the pretraining stage is fundamentally sound and supports the theoretical design of the training phase.
2. Presentation. The overall structure of the paper is clear and the logical progression is relatively natural. The review of background and related work is comprehensive helping readers quickly understand the current research status of large language model compression.
3. Significance. The paper focuses on the memory constrained deployment of large language models on edge devices. This is a research direction of great practical importance and provides new ideas for elastic deployment.
4. Originality. Introducing the alternating direction method of multipliers for structural induction possesses good theoretical novelty. Moreover the proposed adaptive controller can dynamically adjust the rank and sparsity of different network blocks breaking away from the reliance of traditional methods on fixed rules.

Weaknesses:
1. Soundness. The core reasons for the fair rating here are logical inconsistency and unfair experimentation. First the paper emphasizes layer heterogeneity during the training stage but adopts a globally uniform ratio for indiscriminate pruning in the deployment stage parameter allocation strategy which destroys the finely learned structures. Second there is a severe unfairness in the experimental comparisons where the authors compare their models trained in single precision floating point against baseline methods trained in half precision floating point violating the principle of controlling variables. Finally the paper claims to design for resource constrained hardware but lacks real latency or throughput tests.
2. Presentation. The core reason for the fair rating here is the misleading presentation of key information. The paper fails to clearly indicate the precision difference between the baselines and its own method in the core tables hiding the fragile dependence of the algorithm on numerical precision in the appendix. Furthermore the main text repeatedly claims the framework is plug and play but the ablation studies in the appendix reveal that the model is highly sensitive to global hyperparameters making this claim seriously inconsistent with actual fragility.
3. Significance. Although the general direction is important the lack of physical performance evaluation on real edge hardware greatly diminishes its guiding significance for actual industrial deployment and limits its practical value in real world scenarios.
4. Originality. The homomorphic parameter allocation strategy adopted in the deployment stage essentially reverts to the traditional path of globally uniform ratio pruning. Failing to maintain the same level of innovative design in the inference stage as in the training stage weakens the overall original value.

---

> ### Author Rebuttal · Authors · 2026-03-29
>
> Thank you for dedicating your time and expertise to review our submission. Please find our responses below.
> > Logical inconsistency
>
> We do not agree there is a logical inconsistency between two stages. HPA is an efficient approximation that balances pruning efficiency and performance while preserving layer heterogeneity.
>
> As acknowledged by the reviewer, our method respects layer heterogeneity during training, where different blocks learn distinct ranks and sparsities. At deployment, the global coefficients $\phi_L$ and $\phi_S$ only apply proportional scaling to this heterogeneous structure rather than reassigning it, thereby preserving the relative differences across blocks. This differs from methods such as SLTrain and LOST [1,2], which adopt fixed structures in both training and deployment and ignore layer heterogeneity by design. In addition, we explicitly distinguish the structural scale differences between the low-rank and sparse components during parameter allocation and perform compression within each subspace, further reflecting our respect for structural heterogeneity.
>
> We further emphasize that our method is greedy, as the optimum corresponds to solving (8). The non-uniform pruning methods typically incur higher computational cost with limited performance gains. Empirically, on a 1B model, removing over 50% of parameters results in only a 2.4% increase in perplexity, indicating a favorable performance-efficiency trade-off; more sophisticated strategies fall beyond the scope of this work.
>
> From a theoretical perspective, we do not yet provide a rigorous analysis, but this may be related to flatness-based interpretations as in Optimal Brain Damage/Surgeon [3,4], which we leave for future work. We also agree with the reviewer on the suggested baselines; due to time constraints they are not included in the current version, but will be added if possible in the revision, including comparisons with SparseGPT, Wanda, and OWL [5,6,7].
> > Dual standards in precision
>
> We believe the reviewer's concern mainly arises from the presentation rather than the method itself, and does not affect the validity of our conclusions.
>
> First, regarding the concern of "misleading presentation of key information", we note that the current version already reports the difference in training precision between our method and the baselines in both the caption of Table 1 and the main text, without concealment. However, we agree that this information is not sufficiently prominent, and will directly indicate the precision in the main body of the table in the revised version to improve visibility.
>
> Second, regarding the claim of "fragile dependence on numerical precision", our results do not support this. As shown in Table 1 and 4, reducing precision leads to only a slight increase in perplexity (e.g., from 19.70 to 19.73 for the 350M model), which is negligible in practical compression scenarios. Moreover, we do not observe training collapse or compression failure, indicating that the method is not sensitive to precision.
>
> Finally, we report single-precision results in Table 1 to demonstrate the upper-bound performance, as we considered a training-rich, deployment-constrained setting, while we have included half-precision results in the appendix. In the revised version, we will move half-precision results to the main text and single-precision results to the appendix to improve comparability.
> > Lack of physical performance evaluation
>
> Due to space limitations, we provide a detailed discussion on physical performance evaluation in our response to Reviewer C4Yo. Please refer to the corresponding section in that response.
> > Sensitivity of hyperparameters
>
> First, we clarify that the ablation studies on $\rho$, $\Delta \alpha$, and $\Delta \beta$ in the main text and Appendix G are intended to analyze their impact on convergence behavior, rather than hyperparameter sensitivity. The results show that even under variations of approximately one order of magnitude ($\times 10$), performance changes remain limited, indicating strong robustness in practice. Second, as stated in the paper, the same set of hyperparameters can be directly applied to models of different sizes, demonstrating stability under scaling. We will further validate this property on other architectures in future work. Finally, the paper explicitly provided a complete hyperparameter tuning procedure: 1. determine $\rho$ according to (7), 2. leveraging robustness under scaling, perform grid search for the remaining parameters on a smaller model, 3. transfer them to larger models. Since the tuning is conducted on small-scale models, the procedure is simple, scalable, and does not introduce significant overhead. We will further clarify this procedure in the revised version.
>
> [1] Han et al., NeurIPS 2024 [2] Li et al., arXiv 2025 [3] Hassibi et al., ICNN 1993 [4] LeCun et al., Neurips 1989 [5] Frantar et al., ICML 2023 [6] Sun et al., arXiv 2023 [7] Yin et al., arXiv 2023

---

> > ### Author Rebuttal · Reviewer_6Rsb · 2026-04-01
> >
> > Thank you for the thorough rebuttal which effectively addressed my concerns. I will increase my score to 4.

---

> > > ### Author Response · Authors · 2026-04-06
> > >
> > > Thank you for your valuable comments and for taking the time to reconsider our work. We truly appreciate your thoughtful feedback and support. We will incorporate the revisions as outlined in our rebuttal in the updated version.

---

### Official Review · Reviewer_Dauk · 2026-03-13

**Soundness:** 3
**Presentation:** 2
**Significance:** 3
**Originality:** 3
**Overall Recommendation:** 4
**Confidence:** 3

**Summary:**

This paper introduces SALAAD, a plug-and-play pretraining framework that induces sparse and low-rank (SLR) structures in large language models (LLMs) to facilitate memory-efficient deployment. By moving away from fixed schedules, the framework optimizes SLR induction using an augmented Lagrangian approach and a two-stage ADMM-based procedure. This method allows the model to alternate between standard gradient updates and closed-form structural projections. Overall, SALAAD provides a general, architecture‑agnostic approach to training LLMs that can be elastically adapted to diverse deployment memory budgets without retraining.

 Key Technical contributions are

(i) Adaptive I-controller: Automatically regulates sparsity and rank levels across different Transformer blocks, eliminating the need for manual, per-layer hyperparameter tuning.

(ii) Structured Surrogates: Throughout training, the framework maintains a structured surrogate model that tracks the dense weights, ensuring the final output is optimized for compression.

(iii) Homomorphic Parameter Allocation (HPA): A post-hoc strategy that enables a single trained model to scale its capacity elastically to meet diverse memory budgets without requiring retraining.

(iv) Structural Insights: Empirical results on LLaMA-based models (60M to 1B parameters) demonstrate competitive perplexity and significant parameter reduction, with interesting findings showing that even embedding layers naturally adopt SLR structures under this regime.

**Compliance With Llm Reviewing Policy:**

Affirmed.

**Final Justification:**

Thanks to the author for their detailed response which addresses the concerns! I hope they can include these additional results on downstream tasks and efficiency in the manuscript to strengthen their results! I noticed that training time reported as part of author reviewer discussion from additional experiments did not state units. The authors must resolve it. I lean towards accepting the work given its strengths and improvements added during the rebuttal discussion! I retain my score

**Key Questions For Authors:**

Q1: Two‑stage ADMM adaptation uses stochastic minibatch updates for the X‑step and applies only a single SVD‑based structural projection per cycle (Algorithm 1). Could you clarify whether any theoretical guarantees exist or empirical diagnostics beyond bounded reconstruction error ?

Q2: Could you comment on how SALAAD performs on instruction‑tuned models, code‑specialized models, or larger (multi‑billion parameter) checkpoints? Even high‑level results or qualitative observations would help.

Q3: A clearer understanding of overhead helps evaluate real‑world practicality and may improve my significance. While you discuss memory and FLOP overhead in Appendix B, could you provide end‑to‑end measurements such as wall‑clock slowdown, GPU‑hours, and throughput changes across scales? For example: how do the SVD steps (even with J = 1) affect full‑model pretraining cost for 350M and 1B models?

Q4: Understanding failure modes or flexibility of HPA would strengthen assessment. The HPA strategy assumes unit importance scales with magnitude and that all blocks follow a homomorphic scaling pattern (Eq. 9; Figure 6). Have authors  tested scenarios where these assumptions break down such as highly heterogeneous layers or tasks requiring specialized attention blocks? Can HPA allocate differently per module if needed?

**Limitations:**

More discussion points should be included in the writeup:

- Because SALAAD facilitates elastic deployment on edge devices and other low‑resource settings, the authors could discuss possible implications both positive (accessibility, sustainability) and negative (facilitating misuse by lowering resource barriers).

- Although the LM head is noted as non‑SLR‑inducible, broader failure cases e.g., sensitivity to hyperparameters, brittleness under very tight budgets, or mismatch between induced structure and downstream tasks should be discussed explicitly.

**Strengths And Weaknesses:**

### Strengths:

1. The paper presents a well‑grounded ADMM‑based framework with clear mathematical formulation and stable two‑stage optimization behavior. Experiments (Figures 7a–7p) across multiple model sizes show bounded reconstruction error and reliable SLR induction throughout training

2. The paper addresses a major deployment challenge: offering continuous and architecture-preserving model capacity adjustment after pretraining. Unlike prior methods that yield fixed-rank or fixed-sparsity models, SALAAD creates a structured surrogate model that supports flexible parameter budgets via HPA. This allows a single checkpoint to serve a wide range of memory environments from edge devices to GPUs without retraining or architectural changes (Figure 2). This is a meaningful contribution for practitioners deploying LLMs at scale.

3. Where previous methods rely on fixed schedules or architecture‑dependent constraints, SALAAD introduces a unique combination of:
ADMM‑driven training‑time SLR induction, block‑wise adaptive rank/sparsity control via an I‑controller, and post‑hoc homomorphic parameter allocation (HPA) enabling continuous model scaling. The unification of these components into a single plug‑and‑play training framework, without architecture modification, represents meaningful conceptual novelty.

---

### Weaknesses

A. Although ADMM is well‑studied, the paper acknowledges the nonconvexity of neural training and provides no theoretical convergence guarantees for the two‑stage procedure (especially given the stochastic mini‑batch X‑update). The boundedness of $|X − \hat{X}|$ is asserted empirically but not rigorously established for LLM‑scale optimization. Stronger justification could improve soundness.

B. Key sections especially the ADMM derivation, the I‑controller formulation, and HPA strategy are mathematically heavy and may be difficult to parse for readers without optimization background. For instance, the discussion of energy coverage ratios, threshold scheduling, and homomorphic scaling is dense and could benefit from more intuitive explanations or diagrams (e.g., conceptual overviews separate from formulas)

C. SALAAD’s generalization to larger commercial-scale models or instruction‑tuned checkpoints remains unclear. The absence of comparisons in real downstream tasks limits claims about applied impact. There are no results on downstream tasks, and no tests on diverse domains.

D. While the integration of ADMM and adaptive control is well‑designed, several core ideas, training‑time SLR induction (SLTrain, LOST), block‑wise SLR, proximal updates have been explored previously. SALAAD’s novelty comes mainly from combining known techniques and adding adaptive hyperparameter control, rather than offering a fundamentally new theoretical framework. Some readers may view the contributions as evolutionary.

---

> ### Author Rebuttal · Authors · 2026-03-29
>
> Thank you for dedicating your time and expertise to review our submission.
> > Novelty
>
> We agree that related components have been explored in prior work; however, our contribution goes beyond their combination by introducing a training paradigm where SLR structure is adaptively induced through optimization.
>
> For training-time SLR, SLTrain and LOST [1,2] fix the rank and sparsity at initialization, and therefore do not perform structure induction or capture layer-wise heterogeneity. In contrast, SALAAD induces structure during training, allowing different blocks to learn distinct ranks and sparsity. For block-wise SLR, existing approaches are mostly post-hoc compression methods, which incur additional computational cost and do not influence representation learning. Our method instead induces structure during pretraining, directly shaping model parameters and capturing inter-layer differences. Regarding ADMM, we propose a stochastic two-stage implementation that scales to large models. To our knowledge, such approaches have not been systematically applied in large-scale pretraining, and their convergence analysis remains challenging and is left for future work.
> > Theoretical guarantees
>
> We agree that the our work does not provide a rigorous convergence guarantee for the proposed algorithm. Our method can be viewed as an inexact and stochastic variant of multi-block ADMM [3], where the $X$-step only performs stochastic optimization. This setting falls outside the standard assumptions of existing ADMM convergence theory. Extending these results to our setting would require handling both stochastic errors and the nonconvexity of deep NNs. We believe this is a valuable direction but beyond the scope of the current work, and we leave it for future study.
>
> On the empirical side, we further conduct stability analyses under different $K/J$ ratios (i.e., SVD update frequency), and observe that $|X - L - S|$ remains bounded and stable across multiple model scales and a wide range of settings. We will additionally monitor the evolution of the term $\ell_\rho$ during training to provide a more comprehensive diagnostic of stability. These results will be included in the revised version.
> > Instruction-tuned models, etc.
>
> We believe SALAAD is inherently scalable: the same set of hyperparameters can be reused across model scales, and the method is architecture-agnostic, operating directly in weight space, making it naturally applicable to instruction-tuned or code-specialized models. The consistent trends observed across different model sizes further suggest good scalability. In addition, the algorithm supports large-scale parallel training with controllable overhead. We agree that systematic evaluation on larger and specialized models would further strengthen the work and leave this for future study.
> > Downstream tasks
>
> The evaluation on downstream tasks can be found in our response to Reviewer C4Yo.
> > Overhead and inference efficiency
>
> The wall-clock time measurements are provided in our response to Reviewer diEK, while the inference efficiency is discussed in our response to Reviewer C4Yo.
> > HPA strategy
>
> We agree that the current presentation is mathematically heavy. In the revised version, we will improve readability by providing intuitive explanations of energy coverage ratios as the proportion of preserved information in each block and relating them to capacity allocation across layers, and by adding conceptual diagrams for threshold scheduling and homomorphic scaling to illustrate threshold evolution during training and their interaction with the learned SLR structure.
>
> We have not observed clear failure cases of HPA so far. This may be because block-wise heterogeneity is already captured during training, and the subsequent scaling operates on this learned structure, thereby preserving inter-layer differences in practice. We agree that further analysis of the assumptions behind HPA is valuable. In the revised version, we will include comparisons with non-uniform pruning methods such as SparseGPT, Wanda, and OWL [4,5,6], and explore theoretical interpretations from a representation capacity perspective, potentially related to flatness-based analyses such as Optimal Brain Damage/Surgeon [7,8].
> > LM head
>
> We provide a detailed discussion on the SLR behavior of LM head in our response to Reviewer diEK.
> > Positive/Negative implications
>
> SALAAD improves deployment efficiency under resource constraints, which can enhance accessibility and reduce energy consumption, but may also lower the barrier to use and introduce potential misuse risks. We note that our method does not introduce new model capabilities, but only improves efficiency, so the associated risks largely depend on the underlying models.
>
> [1] Han et al., NeurIPS 2024 [2] Li et al., arXiv 2025 [3] Wang et al., SCIC 2018 [4] Frantar et al., ICML 2023 [5] Sun et al., arXiv 2023 [6] Yin et al., arXiv 2023 [7] Hassibi et al., ICNN 1993 [8] LeCun et al., Neurips 1989

---

> > ### Author Rebuttal · Reviewer_Dauk · 2026-04-04
> >
> > Thanks to the author for their detailed response which addresses the concerns! I hope they can include these additional results on downstream tasks and efficiency in the manuscript to strengthen their results!

---

> > > ### Author Response · Authors · 2026-04-06
> > >
> > > Thank you for your kind feedback and for your constructive suggestion.
> > >
> > > We will incorporate the revisions as outlined in our rebuttal in the updated version, including adding additional results on downstream tasks and efficiency to further strengthen our evaluation.
> > >
> > > If you feel that our clarifications and planned revisions have addressed your concerns, we would sincerely appreciate your consideration in your final evaluation.

---

### Official Review · Reviewer_diEK · 2026-03-24

**Soundness:** 3
**Presentation:** 4
**Significance:** 3
**Originality:** 3
**Overall Recommendation:** 5
**Confidence:** 4

**Summary:**

The paper presents a training-time framework that forces weight matrices to learn a dual sparse+LR structure. This allows a single model to be compressed for different objectives w/o retraining. As part of the contribution, the authors introduce an adaptive I-controller and a decoupled ADMM optimization strategy. These mechanisms together avoids the performance drop of post-hoc pruning while remaining compatible with transformer architecture. The result is a highly flexible elastic model that can be scaled differently to fit into different target devices while preserving SOTA accuracy.

**Compliance With Llm Reviewing Policy:**

Affirmed.

**Key Questions For Authors:**

**Q1**: Could you provide a breakdown of the total training time with your approach vs. vanilla training? Does the synchronization introduce a bottleneck in a distributed setting? Also, how does performing SVD more frequently impact the final accuracy?

**Q2**: It was mentioned that the embedding layer exhibits benign SLR behavior, but the LM head does not have this. In doing so, did you experiment with different target compression ratios specifically for the LM head, or did it fail even under mild regularization? Perhaps this could be an interesting ablation study.

**Q3**: In Figure 3, it seems the optimal k stays within 0.6-0.8. Is this dependent on the specific target budget C, or generally stable across the entire spectrum of model capacities?

**Q4**: Finally, setting K=40 seems ad-hoc to me. What happens to the stability of the training loss and perplexity if K is increased/decreased? Does it change the training stability? How did you come up with this value? Do you see variability in this value across different architecture and model sizes?

**Limitations:**

yes

**Strengths And Weaknesses:**

### Strengths

- The use of the augmented Lagrangian and ADMM to solve the SLR decomposition is unique and mathematically rigorous. The two-stage decoupling of training is an interesting way to reformulate heavy optimization work for LLM training. Their empirical results show consistent perplexity gains.

- The paper targets an interesting domain where the gap between different deployment platforms are large. The ability to tune a model's size after it is trained without the need to retrain it from scratch is practical and useful, not only in research settings but also for industry. The additional studies in the paper such as identifying that embedding layers have a benign SLR structure is an interesting insight that could bring a new perspective on how model practitioners think about compressing the first layers of the transformer models.

- While ADMM and LR are not necessarily new, the combination of a controller with weight-space induction is new. As mentioned in the paper, related work such as MatFormer uses nested architectures, but this work achieving the same goal entirely in the weight space, which is unique. Finally, the proposed HPA strategy is an new way to handle global parameter budgets across different types of blocks.

### Weaknesses

Overall I don't have much to say about the weaknesses of the paper but here few that if addressed, it could improve the quality of the paper:

- SVD overhead could become significant as the model size and number of layers grow. The vanilla SVD is notoriously slow on GPUs even if it's just once ever few steps. It would have been great if the authors could provide wall-clock time comparison against standard training. This could show that their method does not slow down pretraining by a large margin.

- The paper focuses primarily on memory and parameter count, however, at the end of the day inference speed is what matters most. Sparse matrices often requires special kernels (or hardware) to actually run faster than dense ones. Without showing actual speedup benchmarks on hardware the significance for limited-memory deployment remains mostly theoretical.

---

> ### Author Rebuttal · Authors · 2026-03-28
>
> Thank you for dedicating your time and expertise to review our submission. Please find our responses below.
> > Breakdown of the total training time
> |Model|Vanilla|ADMM|Sync|Save|
> |-|-|-|-|-|
> |350M|538|89|21|6|
> |1B|2772|1251|347|86|
>
> As shown in the table, we provide a breakdown of training time on 8 H100 80GB GPUs. Compared to vanilla training, the additional overhead mainly comes from ADMM computations, inter-GPU synchronization, and the management of auxiliary variables ($L, S$). In our setup, each GPU processes about 21 matrices. As model size increases, ADMM cost grows accordingly, but can be effectively amortized due to high parallelism across matrices. Synchronization accounts for only a small portion of the total time and does not become a bottleneck, as the algorithm is highly parallel and requires synchronizing only a limited number of variables. We assume sufficient computational resources during training, under which the additional overhead is acceptable. In return, the method enables effective structured compression, providing substantial benefits at deployment.
>
> >  SLR behavior of LM head
>
> In fact, we conducted a series of ablation studies on the LM head, including varying the target compression ratios as well as the strength of structural regularization. We refer to the behavior of the embedding layer as "benign" because moderate regularization can reliably induce a clear SLR decomposition, while preserving the original training dynamics and final model performance.
>
> In contrast, the LM head exhibits fundamentally different behavior. With under mild regularization, we are unable to induce a meaningful SLR structure that supports effective parameter compression. Increasing the regularization strength does enforce the desired structure to some extent, but leads to a significant degradation in model performance. Moreover, we observe that applying structural regularization to the LM head, even at a relatively low level, can noticeably perturb the training dynamics of other layers.
>
> These observations suggest that the LM head differs from the embedding layer in terms of structural compressibility and sensitivity to regularization. To the best of our knowledge, this phenomenon has not been systematically studied in prior work and may warrant further investigation.
>
> We will include additional ablation results in the revised version, covering a broader range of compression ratios and regularization strengths to better support these findings. In future work, we also plan to investigate whether similar behavior arises in other architectures and to better understand the underlying mechanisms.
>
> > Optimal $\kappa$
>
> The reviewer’s concern likely stems from unclear phrasing, which we will clarify in the revised version. As shown in Fig. 3, the optimal allocation ratio $\kappa^{\star}$ varies with model scale (e.g., 0.6-0.7 for 350M and 0.7-0.8 for 1B).
>
> However, for a fixed model scale, $\kappa^{\star}$ remains consistent across different budgets $C$, i.e., it is independent of $C$ and stable across the entire capacity range. Therefore, once the model scale is fixed, the same $\kappa^{\star}$ can be reused across budgets without re-tuning.
>
> > Value of $K$ and SVD frequency
>
> We conducted additional ablation studies and will include them in the revised version. The key finding is that, rather than $K$ or $J$ individually, their ratio $K/J$, i.e., the relative update frequency between the two stages, is the critical factor. Therefore, the effect of more frequent SVD (changing $J$) on accuracy is essentially determined by $K/J$.
>
> With other parameters fixed, $K/J$ controls the strength of structural regularization. A smaller $K/J$ (i.e., more frequent ADMM updates) enforces stronger constraints. In the early stage of training, when $X$ does not yet exhibit an SLR structure, more frequent ADMM updates increase $|X - L - S|$. This accelerates the emergence of a stronger SLR structure, which is beneficial for subsequent compression. However, it also restricts exploration in the weight space, leading to higher perplexity. Despite this, we do not observe noticeable degradation in training stability.
>
> From this perspective, $K/J$ plays a role similar to $\rho$, as both regulate structural strength. In practice, we first choose a reasonable $K/J$ (e.g., 20-40) based on computational cost, and then tune $\rho$ to balance structure and performance.
>
> This design aligns with minimizing hyperparameter tuning: we reduce all effects to the choice of $\rho$, which can be set empirically according to (7). These observations are based on LLaMA-2, and generalization to other architectures remains for future work.
>
> > Inference speed
>
> Due to space limitations, we provide a detailed discussion on inference efficiency in our response to Reviewer C4Yo. Please refer to the corresponding section in that response.

---

> > ### Author Rebuttal · Reviewer_diEK · 2026-04-06
> >
> > Thank you for the rebuttal. Please ensure including these additional results and clarifications in the revised manuscript. I maintain my positive score.

---

> > > ### Author Response · Authors · 2026-04-07
> > >
> > > Thank you for your positive feedback and support. We will incorporate the additional results and clarifications in the revised manuscript.

---

### Decision · Program_Chairs · 2026-04-30

**Decision:**

Accept (regular)

**Comment:**

This paper proposes a very nice formulation to encourage Sparsity and Low Ranked ness of weight matrices using ADMM for LLMs. The writing, math and experiments are really clean and all the reviewers agree that it adds value to the community. While I understand the compute constraint, I strongly urge the authors to incorporate neccesary discussion on the suggestions from the Reviewers. With relibale scaling laws and scale ups, I could see this work having strong impact down the line.